# BUILDING SIMULATION ENVIRONMENTS FOR COMPUTATIONAL ORGANIZATIONAL DESIGN

## ABSTRACT

Organizational success depends less on individual brilliance than on how teams are structured, coordinated, and adapted. Yet organizational design remains a grand challenge in computational science, and machine learning lacks tools to address it. We introduce the **Organizational Design Problem (ODP)**: learning a *management policy* that configures team composition, communication, and autonomy to achieve multi-objective goals under structural constraints.

A main obstacle to developing machine learning for the ODP is the lack of suitable **Organizational Simulation Environments (OSEs)** in which such policies can be learned and evaluated. While organizational design is a general task as organizations are a universal feature of social and economic life, each organization is unique in its purpose, internal constraints, and external surroundings. Acknowledging this specificity, we propose an OSE blueprint: it defines the core components shared by all organizations while allowing adaptation to diverse contexts. In this framework, fixed LLM agents simulate realistic human roles and communicate via natural language within a mechanistic, temporally grounded simulation.

Applying this blueprint, we present the **Clinical Trial OSE**, which captures the high-stakes, multi-stakeholder process of drug development. Using this environment to benchmark pre-trained LLMs, we show that they can guide organizations to successfully complete trial programs. Although current models remain less efficient than humans, our study opens the path toward specialized models that could one day outperform humans in systematically solving the Organizational Design Problem.

## 1 INTRODUCTION

The intricate dynamics of human organizations, from research lab to multinational corporations, present a formidable optimization challenge. Organizational success is often driven not by individual brilliance, but by the emergent properties of communication, collaboration, and structure. Traditional analytical methods struggle to capture these complex and often unpredictable interactions. Meanwhile, machine learning lacks a dedicated paradigm for optimizing the fundamental elements of organizational design. How should team schedules be structured to achieve different objectives? What is the impact of communication policies on costly delays? How robust is an organization to individual mistakes? These remain foundational yet unsolved questions in organizational science.

**What is an Organization?** Following Barnard's classic view, a formal organization exists when people communicate, contribute action, and pursue a common purpose (Barnard, 1938). We treat organizations as three linked elements. *(i) Purpose.* Purpose justifies existence and shapes design. It is typically multi-dimensional: a vector of negotiated objectives (e.g., production, quality, budget, safety) pursued under constraints (Cyert & March, 1963; Scott & Davis, 2015). *(ii) Members.* Individuals contribute functional roles (skills, information, resources) and bring preferences and incentives; effectiveness requires aligning personal inducements with collective purpose (Katz & Kahn, 2015; Parsons, 2013; Barnard, 1938). *(iii) Cooperative system.* Interdependent tasks are coordinated through standards, planning/scheduling, or mutual adjustment, (Thompson, 1967; Mintzberg, 1979).

Moreover, organizations do not operate in a vacuum. They are embedded in environments populated by other actors and conditions (Katz & Kahn, 2015; Scott & Davis, 2015). They depend on their

environment for resources and are constrained by it. Reliance on external parties for capital, materials, data, or approvals creates power relations that must be managed (Pfeffer & Salancik, 2015). Consequently, environmental conditions shape organizational design (Lawrence & Lorsch, 1967).

**What does it mean to optimize an organization?** In our framework, optimizing an organization means configuring its internal state: how members and cooperative systems are arranged to achieve its purpose. Concretely, this entails maximizing multiple (often competing) objectives that define the organization's purpose, subject to constraints and uncertainty arising from a dynamic, stochastic environment. Effective organizational design therefore requires *robustness*: the capacity to adapt structure and coordination to environmental change while sustaining purpose. We formalize this optimization problem along three design parameters:

1. **Composition of elements** – Selecting the right actors based on task demands. Seeking highly capable actors may be costly, while insufficient diversity or quantity can stall progress.

2. **Communication policy** – Designing efficient protocols to coordinate actors and ensure effective information flow tailored to the task.

3. **Autonomy** – Balancing centralized versus decentralized control by adjusting autonomy and aligning incentives to steer behavior. This collective capacity is achieved at the cost of limiting the degrees of freedom of its members.

**Organizational Design Problem.** Building on prior work in computational organizational design (Carley, 1994), we formalize a machine learning task: learning a *management policy* that jointly controls the three organizational levers defined above to optimize system-level performance while holding individual agent capabilities fixed (see section 2.1). With this framing, we aim to bring new analysis tools to support the systematic design of organizational structure.

**Why has this not been addressed before?** Despite the importance of organisational design, machine learning has not tackled this learning task due to the lack of suitable simulation environments (section 5). Training and testing policies on real organisations is ethically, financially, and operationally infeasible; meaningful progress requires simulations that capture both agents' decisions and their external context. Just as physics relies on simulated environments to study galaxies or molecules, organisational science requires high-fidelity simulations to understand and optimise complex human systems. Our framework provides such an environment, enabling controlled experiments on structure, communication, and incentives that would be impractical in vivo.

Recent advances in LLMs make this possible by enabling human-like agents capable of natural-language coordination and complex task reasoning (Zhou et al., 2024b; Park et al., 2023), and can perform office-style work (Xu et al., 2025; Boisvert et al., 2025). However, existing LLM simulations are rarely grounded in non-LLM surroundings, limiting their evaluative power. We close this gap by pairing **LLM-based agents** with **domain-grounded mechanistic simulations**, providing a testbed in which management policies can be learned and assessed under realistic constraints.

**Why a reinforcement learning environment?** A learned *management policy* is queried at every time step, enabling continuous, adaptive reconfiguration of an organization, a granularity that cannot be matched by expert-designed plans. This aligns with *contingency theory* (Dobbin, 1998), which holds that optimal structures must evolve with changing internal and external conditions. Reinforcement learning provides a natural framework for such adaptive, long-horizon control. However, our OSE differs from existing multi-agent environments in two ways: (i) it optimizes organization-level levers: agent selection, communication, and collaboration; rather than individual policies; and (ii) its hybrid nature: coupling LLM-based agents with mechanistic models.

Our work aims to bridge the gap between traditional reinforcement learning and LLM-based agents. Currently, the available environments to explore this intersection are mostly OS and web-based tasks (Xie et al., 2024; Zhou et al., 2024a). By creating an environment that faithfully captures the dynamics of real-world organisations, we contribute to extending this line of work to strategic, high-stakes domains requiring reasoning, memory, and language understanding.

**Use case: Clinical trials**. To demonstrate the power and utility of our benchmark, we instantiate it in one of the most complex, high-stakes domains: clinical trials. The development of a new drug is a decade-long, multi-billion dollar process where over 90% of initiatives fail, often due to logistical inefficiencies and poor strategic planning rather than scientific shortcomings. A clinical trial

serves as a perfect microcosm of organizational complexity (see simulation reports in section C.2), involving diverse stakeholders, strict regulatory constraints, and critical deadlines. By modeling the key actors, resources, and tasks within a trial, we create the first environment of its kind, enabling researchers to explore how different organizational structures impact drug development.

In summary, our contributions are:

1. The **Organizational Design Problem (ODP)** is defined as learning a management policy over (i) actor composition, (ii) communication policies, and (iii) agent autonomy to optimize multiple objectives under constraints. We cast the ODP as a multi-objective POMDP, providing a rigorous basis for discovering and evaluating organizational structures.

2. We introduce **Organizational Simulation Environments (OSEs)**: a blueprint that couples domain-specific mechanistic models with fixed LLM agents that communicate in natural language within a discrete-time simulation. The blueprint specifies core components and engineering patterns, enabling the first environments for training and evaluating LLM agents on the ODP across domains.

3. We release an open-source Phase II **Clinical Trial OSE** modeling up to 25 different actors and 8 drugs across 5 scenarios, with elements such as patients, adverse events, clinical studies, biomarkers, PK/PD, and regulatory constraints toward Phase III approval. This benchmark enables controlled studies of how organizational structure affects the speed, cost, and success of drug development.

4. We benchmark baseline management policies, including a large range of pretrained LLMs, on the **Clinical Trial OSE**, yielding insights for developing stronger management policies.

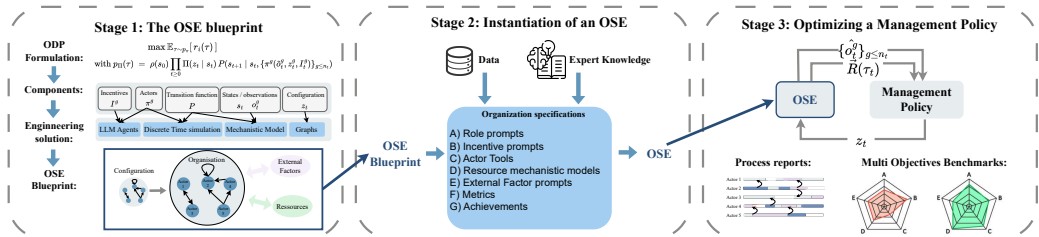

Figure 1: **Roadmap for Computational Organizational Design.** This paper introduces a blueprint for Organizational Simulation Environments (OSE) in section 2, presents a concrete instantiation in section 3, and benchmarks baseline management policies in section 4.

## 2 COMPUTATIONAL ORGANIZATIONAL DESIGN

### 2.1 DEFINITION OF THE ORGANIZATIONAL DESIGN PROBLEM (ODP)

**Single-agent setting.** As the Organizational Design Problem can naturally be cast as a multi-agent reinforcement learning problem, we first model it as a decentralized partially observable Markov decision process (Dec-POMDP) Oliehoek et al. (2016) for a set of $n$ agents, defined by the tuple $\langle O^n, S, A^n, P, R \rangle$, where $O^n$ is the set of joint observations, $S$ is the state space, $A^n$ is the set of joint actions, $P$ is the stochastic transition function, and $R$ is the reward function. However, as in real organizations, operational actors cannot be directly controlled; we assume that their policies are fixed and that only the management policy can be chosen. Under this assumption, the actors can be absorbed into the transition dynamics of an induced decision problem faced by the manager, which reduces the ODP to a **Multi-Objective Partially Observable Markov Decision Process (MOPOMDP)** defined by the tuple $\langle \Theta, S, \mathcal{Z}, P, \vec{R} \rangle$. Here, we rename the management policy observation space as $\Theta$, its action space as $\mathcal{Z}$, and $\vec{R}$ denotes a vectorial reward. We retain the Dec-POMDP notation to describe low-level actor interactions. Additional rationale for the reduction from Dec-POMDP to MOPOMDP is provided in section 5.

**Actors.** An actor $g$ has a fixed policy $\pi_g(a_t^g \mid \tilde{o}_t^g, z_t^g, I_t^g)$ which at time $t$ defines a distribution over action $a_t^g$ from the actor action space $A$ conditioned on the aggregated observations $\tilde{o}_t^g$, a configuration $z_t^g$, and an incentive $I_t^g$. Since only a fraction of the environment is accessible to the organization and an even smaller portion observable by the actor, each actor receives a partial observation $o_t^g$ in $O$. To relax the Markov assumption, we aggregate past observations into $\tilde{o}_t^g =$

$\sum_{k=T_0}^{t} o_k^g$, where $T_0$ is the time the actor joined the organization. The policy of each actor can be modulated by configuration $z_t^g$ from the configuration space $Z$ and explained by incentive $I_t^g$ from the incentive space $\mathcal{I}$. We consider temporally extended actions over multiple time steps modeled by a blocking state such that, if $a_t^g$ has duration $T$, then

$$\forall k \in [t+1, t+T], \forall \tilde{o}_k^g, z_k^g, I_k^g \in O \times K \times \mathcal{I}, \quad \pi_g(a_k^g = a_t^g \mid \tilde{o}_k^g, z_k^g, I_k^g) = 1$$

**Organizations.** From the management policy's perspective, its own observation $\theta_t$ from $\Theta$ at time $t$ is defined as the set of $n_t$ actors: $\theta_t = \{\pi_g(\tilde{o}_t^g, \cdot, ?)\}_{g \le n_t}$, which embeds both actors' experience (aggregated observations) but keeps their incentives hidden. Giving the management policy access to actors' observations at every time step is known as the synchrony assumption (Messias et al., 2013). Here, we adopt a weak form of this assumption, since only the management policy, and not every actor, has access to each actor's observation. The management policy $\Pi$, from the policy space $\Psi$, therefore chooses the organizational size $n_t$ and the configuration vector $z_t = (z_t^1, z_t^2, \ldots, z_t^{n_t})$ to pursue its objectives. Thus, the action space $\mathcal{Z}$ is $Z^*$ the set of finite sequences of configurations.

**Transition Function.** The management policy influences the environment only indirectly, by configuring the actors' behavior. Only the actors $\{\pi^g\}_{g \le n_t}$ can directly interact with the environment. From the perspective of management, the stochastic transition function $P$ can be expressed as

$$\begin{aligned} P(s_{t+1}|s_t, \{a_t^g\}_{g \le n_t}) &= P(.|s_t, \{\pi^g(\tilde{o}_t^g, z_t^g, I_t^g)\}_{g \le n_t}) \\ &= P(.|s_t, z_t) \ (\pi_g \text{ is fixed and } \tilde{o}_t^g, I_t^g \text{ are included in } s_t). \end{aligned}$$

**Vectorial Reward Function.** To capture the competing goals inherent in any complex organization, the vectorial reward function $\vec{R}$ maps a trajectory $\tau \in \mathcal{T}$ to a reward vector of dimension $d$.

$$\vec{R} : \mathcal{T} \to \mathbb{R}^d; \qquad \vec{R}(\tau_t) = (r_t^{(1)}, r_t^{(2)}, \ldots, r_t^{(d)})$$

Each component represents a distinct objective and is typically sparse and event-driven. Thus, the management policy goal is not to find a single "optimal" policy, but rather to discover the **Pareto frontier**: the set of policies where no single objective can be improved without degrading another.

A trajectory is $\tau = (s_0, z_0, s_1, z_1, \ldots)$ with distribution under the management policy $\Pi \in \Psi$ and initial state distribution $\rho$ is given by $p_\Pi(\tau) = \rho(s_0) \prod_{t \ge 0} \Pi(z_t \mid s_t) P(s_{t+1} \mid s_t, z_t)$.

For a policy $\Pi \in \Psi$, the expected objectives are

$$J(\Pi) = \mathbb{E}_{\tau \sim p_\Pi}[\vec{R}(\tau)] = \big(J_1(\Pi), \ldots, J_m(\Pi)\big), \qquad J_i(\Pi) = \mathbb{E}_{\tau \sim p_\pi}[r_i(\tau)].$$

For two management policies $\Pi, \Pi'$ in $\Psi$, we define $J(\Pi) \succeq J(\Pi') \iff J_i(\Pi) \ge J_i(\Pi') \quad \forall i$,

A management policy $\Pi^\star$ is **Pareto-optimal** if $\nexists \Pi \in \Psi : J(\Pi) \succ J(\Pi^\star)$. The set of all such policies is $\Psi^\star = \big\{\Pi \in \Psi \mid \nexists \Pi' \in \Psi : J(\Pi') \succ J(\Pi)\big\}$.

The **Pareto frontier** in objective space is $\mathcal{F} = \big\{J(\pi) \in \mathbb{R}^m \mid \Pi \in \Psi^\star\big\}$. Thus, the multi-objective reinforcement learning problem is to characterize $\Psi^\star$ and its image $\mathcal{F}$.

## 2.2 Organizational Simulation Environment (OSE)

The OSE provides a framework to train and benchmark management policies. It instantiates all components of the formalism introduced above, except the management policy itself. In this subsection, we describe how to implement these components in practice for any domain.

### 2.2.1 State Space

The environment must support diverse, interdependent tasks that can only be solved through collaboration among multiple actors to provide realistic challenges for management policies. Since organizations are fundamentally human-centered, we aim to replicate human decision-making for these actors. The state captures both the organization and its surrounding environment, which we partition into two components with distinct simulation requirements (see fig. 2).

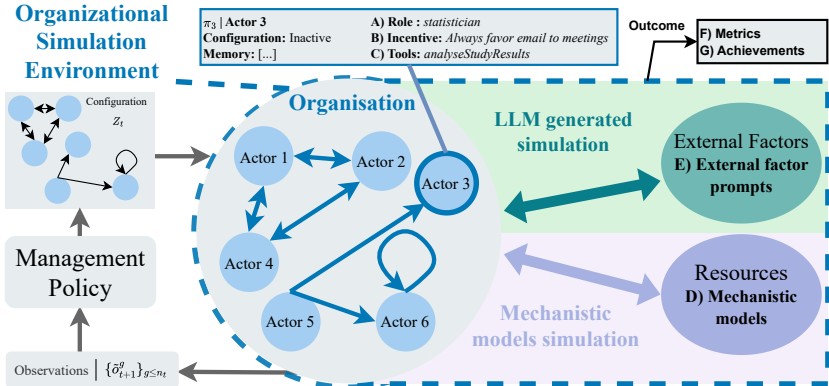

Figure 2: **Overview of an OSE.** At each time step, the management policy selects an organizational configuration $Z_t$. In this configuration, actors reason, communicate, and interact with resources and external factors. At the end of the step, their observations are fed back to the management policy.

**LLM generation.** We rely on LLMs to simulate both the actors' policies $\pi_g$ and the *external factors*. External factors denote entities outside organizational control, such as customers, providers, or competitors, with which the organization must interact. Since these interactions are at least partly human in nature, LLM simulation provides greater realism.

**Discrete-time simulation.** Non-human, quantitative *resources* should be modeled mechanistically in discrete time. This includes variables directly tied to objectives $\vec{\mathbb{R}}$, where LLM noise would be undesirable (e.g., time, money, machines). Discrete-time rules also govern communication and coordination among actors, ensuring synchronization across the simulation.

### 2.2.2 TRANSITION FUNCTION AND TIME

The OSE evolves according to a stochastic transition kernel $P(s_{t+1} \mid s_t, z_t)$ that integrates both organizational dynamics and temporal constraints. Time plays a critical role in shaping organizational efficiency. Each action incurs a time cost, and each actor should be single-threaded to reflect cognitive limitations. Thus, trade-offs between reasoning and communication, and choices of communication partners, become integral to structural optimization (as shown in fig. 5). OSEs are therefore fundamentally *event-driven*: actor actions, resources, and external factors consume heterogeneous amounts of the "time" resource and schedule events that influence future $s_t$. The transition kernel is applied at each time tick of the internal clock of the simulation. Discrete-time control enables transparent measurement and robust regulation of stochastic LLM generation throughout the rollout, mitigating hallucination drift and preserving a precise record of the numerous LLM calls.

### 2.2.3 ACTOR MODEL

To simulate actors that behave as closely as possible to humans, we model each actor with an LLM. Hence, an actor defined by the conditional distribution $\pi_g(a_t^g \mid \tilde{o}_t^g, z_t^g, I_t^g)$ is instantiated as an LLM whose prompt consists of the system prompt associated with its role $\gamma_g$ concatenated with the actor's parameters $(\tilde{o}_g, z_g, I_g)$. This setup leverages the few-shot reasoning abilities of pre-trained LLMs and offers several advantages. First, it enables natural language observations and actions, which are both expressive and interpretable, and allow for communication in human-like language. Second, it provides flexibility in defining and modifying both configurations and incentives. However, this approach also raises several technical challenges:

**Memory.** While aggregated observations may be arbitrarily long, LLMs have limited context windows. A memory mechanism is needed to compress older observations into compact representations.

**Tools.** While LLMs excel at generating natural language, they may hallucinate and are less reliable for simulating non-linguistic complex actions. To address this, such actions are implemented via discrete-time simulation and exposed to the LLM as callable tools (Team, 2024).

**Simplified assumptions.** Real organizations are composed of heterogeneous individuals with diverse personalities, skills and experience. To reproduce this diversity in a controlled manner, we make the following assumptions: (i) Each actor $g$ has a fixed role $\gamma_g$ from a finite set of roles $\Gamma$, which determines its system prompt and the set of available tools. A role $\gamma$ restricts the actor's actions and observations to subsets $A_\gamma \subseteq A$ and $O_\gamma \subseteq O$ of the global action and observation spaces, so that these spaces can be written as unions of role-specific subspaces:

$$A = \bigcup_{\gamma \in \Gamma} A_\gamma, \qquad O = \bigcup_{\gamma \in \Gamma} O_\gamma.$$

(ii) Each actor has a fixed incentive $I_g$ from a finite set of incentives $\mathcal{I}$, that modulate its behavior, introduce misalignment with organizational goals, and remain hidden from the management policy. These incentives do not change over time ($\forall t > 0, I_t^g = I_g$). When choosing to expand the organization, the management policy can select the role of new actors, but their incentives are randomly assigned. These assumptions allow us to generate diverse agents using a fixed library of human-defined roles and incentives.

## 2.3 CONFIGURATION SPACE

In this version of the OSE, we define the configuration space $Z$ as a set of mutually exclusive states: **Reasoning**, **Communicating Asynchronously**, **Communicating Synchronously**, or **Inactive**.

**Reasoning.** The actor (i.e. an LLM) generates internal reasoning traces, task-related deliverables, or tool calls. These outputs remain private and are not observed by the other actors.

**Communicating Asynchronously.** The actor produces messages or reasoning traces directed to a group of recipients. Messages are appended to the recipients' aggregated observations.

**Communicating Synchronously.** The actor engages in real-time interaction with a designated group of peers. All participants must also be in the synchronous state with the same group, and all generated messages are shared among them, allowing for immediate responses.

**Inactive.** The actor produces no output.

Each configuration can be represented as a mixed graph: **1)** Nodes represent actors. **2)** Self-loops denote reasoning. **3)** Directed edges denote asynchronous communication. **4)** Bi-directional edges denote synchronous communication. **5)** Nodes without outgoing edges are inactive.

State exclusivity requires each node to have at most one type of outgoing edge (self-loop, directional or bi-directional). Valid synchronous communication further requires that the corresponding connected component forms a fully connected subgraph. Two opposing directed edges correspond to two distinct asynchronous communications, while a single bi-directional edge represents one synchronous interaction. See fig. 1 and fig. 2.

## 3 INSTANTIATING AN OSE : A CLINICAL TRIAL CASE STUDY

Drug discovery and development is a lengthy, costly, and high-risk process, often taking 10–15 years and costing over \$2 billion per approved drug (Sun et al., 2022). Despite this investment, nearly 90% of clinical drug development efforts fail. Optimizing the organizational processes involved in clinical trials is thus critical to maximize the efficient use of time and resources. Poor resource management directly affects the commercial viability of a drug. For instance, delays reduce the effective patent window (Williams, 2017), discouraging full trial development, while mismanagement increases costs and raises the profitability threshold, leading to premature trial termination (Sertkaya & Franz, 2022). Consequently, failures in clinical trials are often attributed to management inefficiencies and lack of strategic planning (Sadoon et al., 2023; Sun et al., 2022).

Clinical development proceeds in three phases. Phase I is highly standardized; Phase III largely scales designs established in Phase II. Phase II, by contrast, offers the richest design space yet suffers high attrition (about 66% rejection) (Torres-Saavedra & Winter, 2022). Accordingly, our case study instantiates an OSE for Phase II clinical trial programs by specifying seven elements: **A) Role-definition prompts, B) Incentive prompts, C) Actor tools, D) Mechanistic models, E) External-factor prompts, F) Metrics, G) Achievements**.

## 3.1 INSTANTIATING THE ORGANIZATION

As detailed in section 2.2.3, each actor is specified by **A) a role prompt**. In the Clinical Trial OSE, we consider four roles: Investigator, Clinical Program Lead, Statistician, and Regulatory Lead (system prompts are listed in section D.5). Each role has access to a distinct set of **C) tools** (e.g., `DesignSingleArmStudy`, `ApproveClinicalStudy`, `AnalyseStudyResults`, `SubmitApplicationPhaseIII`) that interface with organizational resources and external factors. Tools incur heterogeneous time costs (e.g., `AnalyseStudyResults` requires 4 hours for a single-arm study and 8 hours for a comparative study) and may fail if used incorrectly (e.g., due to invalid parameters or timing). A complete catalog of tools is provided in table 6.

To further diversify behavior, we define nine **B) incentive prompts** that assign each actor a distinct identity. Some incentives are role-specific, such as the Investigator's risk aversion: "*The actor is very cautious about side-effects. They do not tolerate any mild adverse events.*" Others are generic, such as laconic communication: "*The actor always keeps communication to a minimum, sending the shortest possible emails.*" Combining roles with incentives yields up to 25 unique actor profiles, sampled at hire by the management policy. The full set of incentives and experiments illustrating their effects is summarized in table 8.

## 3.2 INSTANTIATING THE ORGANIZATION'S SURROUNDINGS

The organization's surroundings comprise two elements: resources and external factors. Resources are simulated with **D) mechanistic models** grounded in medical and biological knowledge. Table 1 lists the resources defined for the Clinical Trial OSE and their associated models.

External factors are actors outside the organization, instantiated as LLM agents via **E) external factor prompts**. In the Clinical Trial OSE, a key external factor is the *Regulatory Agency*, which interacts with the *Regulatory Lead* through the tool `submitApplicationPhaseIII` to guide the Phase III application (prompt in section D).

These elements, together with the organization, are integrated into a discrete-time simulation using SimPy (Zinoviev, 2024), a Python-based discrete event simulation framework.

Table 1: **Resources and their associated mechanistic models**

| **Resources** | Drug | Biomarker | Adverse Events | Patient | Single Arm Study | Comparative Study |
|---|---|---|---|---|---|---|
| **Mechanistic Models** | Pharmacokinetics | Dose-response | Risk models | Phenotype | Dosage, Recruitment, Admin | Dosage, Recruitment, Standard of care, Admin |

## 3.3 INSTANTIATING THE VECTORIAL REWARD

The vectorial reward is instantiated through **F) metrics**. A special metric is the **Completion** objective, which quantifies how far a management policy progresses within a simulated scenario. To evaluate completion, we introduce **G) Achievements**: verifiable meta-tasks that require coordinated actions from one or multiple actors. Clinical Trial OSE's achievements are provided in section D.

Beyond completion, the vectorial reward for the Clinical Trial OSE combines six additional metrics, yielding a seven-dimensional evaluation. The **Correct outcome** metric measures the percentage of trials reaching the expected result. In our environment, a Phase II trial can terminate in one of three ways: interruption due to safety or efficacy concerns relative to the standard of care, transition to Phase III, or expiration of the time limit without completion. Each drug is associated with one of the first two outcomes as its expected result (table 7). Additional metrics capture the cost of running the organization. The **Total time** records the simulated duration until episode termination, since longer trials reduce the exploitation window, while the **Worked time** captures cumulative working hours across all actors. Secondary cost factors include the **Study completed count** and the **Patient hired count**. Finally, patient well-being is assessed through the **Adverse event count**.

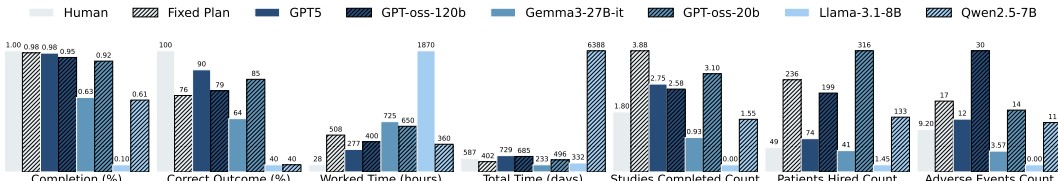

Figure 3: **Comparison between Human, Fixed Plan and LLM policies.**

# 4 EXPERIMENTS AND RESULTS

## 4.1 BENCHMARK

**Baselines.** fig. 3 compares three types of policies on the **Clinical Trial OSE**, reporting performance across the vectorial reward metrics. The policies are: **1. Human**: a domain expert familiar with clinical trials and the environment acts as the management policy. **2. Fixed Plan**: a predetermined schedule that cycles through meetings, reasoning, and email exchanges, inspired by structured routines commonly used in organizational management (Orlikowski & Yates, 2002). **3. LLM**: pretrained LLMs used as management policies. Detailed experimental details in section B. In all experiments, environment actors are powered by Qwen2.5-7B.

**Results.** Fixed Plan and GPT-5, the best-performing LLM policy, achieve completion scores comparable to the Human policy but reach the correct outcome less frequently. Both require substantially more resources: Fixed Plan uses 18× more working hours, while GPT-5 requires 10× more hours and extends trial duration by 20%. They also rely on additional clinical studies, increasing adverse events and patient recruitment. GPT-5, however, is more efficient than Fixed Plan, using fewer hours and progressing faster, highlighting potential benefits of LLM-based policies. By contrast, smaller LLMs such as Qwen2.5-7B and Llama-3.1-8B perform poorly, with Llama completing only 10% of trials and both achieving just 40% correct outcomes. Given the variability in drug difficulty, per-scenario results with standard deviations are reported in section C.1.

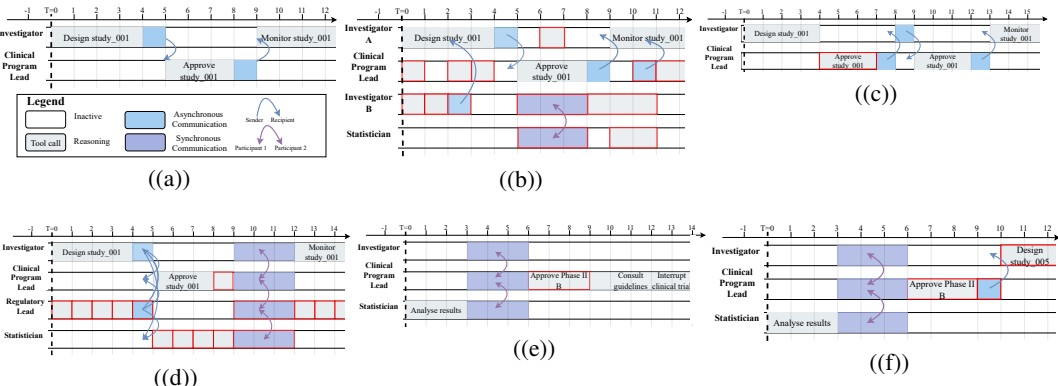

Figure 4: **Simulation report extracts illustrating recurrent inefficiencies across policies.** Full reports in section C.2. **(a)** Human policy: study designed, approved, and monitored with timely communication. **(b)** LLM inefficiencies: noisy active states. **(c)** LLM inefficiencies: missed communication. **(d)** Fixed Plan inefficiency: rigid, poorly timed actions. **(e)** Human correction of actor error in adverse event handling. **(f)** The LLM fails to detect the error and proceeds incorrectly.

**Modes of failure.** The Clinical Trial OSE produces a full report after each simulation episode (section C.2), which helps us understand why some policies are less efficient. fig. 4 shows typical cases: **(a)** To start a clinical study, the Human policy follows a clear sequence: the study is designed, then approved, and finally monitored. After each reasoning step, the human makes sure to communicate by email. This structured process allows the trial to progress smoothly. **(b)** In comparison, LLMs sometimes add extra "active" states that do not help the trial. In this case, actors keep reasoning and sending messages without moving the study forward. These noisy steps waste working time, delay progress and are likely to confuse the actors. This suggests that management policies must learn

when actors should remain inactive rather than defaulting to constant action, in order to avoid noisy over-activity that wastes resources and creates downstream errors. **(c)** Another frequent issue is that LLMs skip important communication steps. Here, a message expected at $t = 4$ is missing, which later forces the actors to repair the situation. This shows how small mistakes in communication can create larger problems downstream, so the timing and structure of communication between actors are critical and must be optimized to avoid large coordination failures and unnecessary delays. **(d)** The Fixed Plan policy cannot adapt. It keeps rotating through predefined actions, even when they are not appropriate. For example, reasoning happens when communication would be needed, and vice versa. This rigid cycle produces many useless working hours and illustrates that fixed, non-adaptive policies are inefficient, motivating state-dependent management policies that can adapt to the dynamics of the organization. **(e)** In a Phase IIb transition, the Clinical Program Lead receives results showing an increase in severe adverse events. The correct choice is to stop the trial, but the actor fails to do so. The Human policy forces the actor into reasoning mode to detect and fix the mistake, preventing unsafe continuation. **(f)** The same situation as (e) occurs with GPT-5, but here the mistake is not corrected. GPT-5 shows reduced robustness to actor mistakes and even starts a new study, indicating that domain-specific field knowledge is essential and cannot be acquired efficiently from interaction with the environment alone. Together, (e) and (f) indicate that organizations must be designed to be robust to local actor mistakes, with mechanisms that detect and correct errors rather than allowing them to propagate through the system.

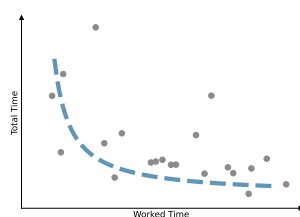

Figure 5: **Pareto Frontier**

**Pareto Frontier.** The ODP is multi-objective: we seek the Pareto frontier of $\vec{R}$. To illustrate this, we extend the Fixed Plan into a family of management policies parameterized by *Length* (size of the reasoning window) and *Stride* (number of inactive steps within windows). Varying them produces different team schedules whose **Worked Time** and **Total Time** performance (fig. 5) exhibits a Pareto trade-off (blue dashed line); no single policy minimizes both instead; one navigates the frontier to select the schedule that prioritizes the objective at hand.

**Cost and fidelity of the Clinical Trial OSE.** Running one episode (a full clinical trial) costs $< \$1$ on average (see table 5), covering environment and execution of the GPT-5 management policy. The policy contributes $\sim 80\%$ of cost, reflecting GPT-5's higher API price relative to Qwen. We define hallucinations as generating factually incorrect or nonsensical content (Xu et al., 2024c) (e.g., wrong study IDs, fabricated results). Actor logs show a $15\%$ hallucination rate, with $> 90\%$ concentrated in $10\%$ of episodes, often triggering cascading errors. An analysis of triggers indicates that hallucinations concern only study manipulation: $17\%$ arise from hallucinating the wrong study identifier, $33\%$ from inventing a new study, and $50\%$ from hallucinating results or reviews of an existing study that has not yet been run. In addition, in our experiments, LLM actors exhibit a relatively high failed tool-call rate of $47\%$, consistent with known limitations of current models, especially smaller 7B architectures. We mitigate these failures using a reflection framework. Moreover, the human policy operating with the same LLM actors achieves substantially higher performance, indicating that our main conclusions are not driven by systematic tool-usage errors.

## 5 RELATED WORK

This section summarizes the extended discussion in section A, which clarifies the positioning of the ODP task, compares it with existing environments, and reviews ML for clinical trials.

Multi-Agent Reinforcement Learning (MARL) and its hierarchical variants, such as hierarchical Dec-POMDP (Oliehoek et al., 2016), study coordination, communication, and scheduling for task allocation and policy learning (Kim et al., 2019; Jiang & Lu, 2018). Yet, they rarely combine features central to real organizations: natural-language interaction, partial goal alignment, fixed heterogeneous agents' policies, non-fixed agent set, mechanistic grounding, and long-horizon optimization. LLM-based agentic frameworks model teamwork, negotiation, and game-theoretic behavior (Li et al., 2023b; Hua et al., 2024a; Fontana et al., 2025) and systems like ChatDev, MetaGPT, and AutoGen improve task decomposition and collaboration (Qian et al.; Hong et al.; Wu et al.), but they typically optimize task performance within purpose-built agents rather than organizational de-

sign. Existing benchmarks reflect this split: MARL suites (e.g., PettingZoo, StarCraft) lack human-like language agents, while social simulations (e.g., Sotopia, WarAgent, Smallville) emphasize dialogue without mechanistic environments. Our work formulates organizational design as a learning problem that optimizes structure and communication among fixed LLM-based roles under realistic constraints, integrating natural language, mechanistic grounding, mixed alignment, and temporal modeling to enable systematic benchmarking.

Machine learning for clinical trials spans evidence synthesis, design, cohort construction, matching, outcome prediction, and auxiliary tasks such as summarization and writing, but prior simulations tend to emphasize biological or statistical models or narrow LLM tasks like QA and trial prediction. We instead introduce an organizational-level simulation of clinical trials that models interacting agents and tasks over time, uses LLMs to emulate actor decisions, and exposes an RL interface for learning management policies. By operating at system scale with modular integration and configurable organizational policies, our environment enables rigorous study of clinical trial management as an optimization problem, bridging LLM-based social simulation with mechanistically grounded, long-horizon organizational design.

## 6 DISCUSSION

**Limitations.** While we propose a blueprint to build OSEs, filling the required components with data and expert knowledge still demands substantial modeling effort to design mechanistic models and coordinate their interactions. This engineering work is essential, as the quality of the learned management policies depends directly on the fidelity of the simulation. For instance, the Clinical Trial OSE was developed in close collaboration with academic and industry experts in clinical trials, who reviewed and validated the simulator.

**Broad Impact.** This work enables the development of management policies at the intersection of Reinforcement Learning and LLM agents. Our results (section 4) already suggest actionable design principles and open directions. Beyond management policies, the OSE blueprint can seed a broader family of environments across domains well beyond clinical trials, e.g., companies, government agencies, educational institutions, and charities, reflecting the generality of organizational structure. Developing such environments can advance multiple areas in parallel, including multi-agent RL, LLM-based agents, inter-agent communication, memory modeling, and both mechanistic and language-based simulation.

**Future of Organizational Simulation Environments.** The roadmap for the Clinical Trial OSE and the general OSE blueprint includes enriching the environment with new actor types, additional drugs, and coverage of earlier (Phase I) and later (Phase III) trial stages. Another direction is enhancing agent capabilities, for example, by integrating retrieval-augmented generation (RAG) systems or fine-tuning role-specific agents. Most importantly, methods to automatically generate OSEs from data or expert interaction would address current limitations and enable scalable development of domain-specific environments. .

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

## A  DETAILED RELATED WORK

### A.1  ORGANIZATIONAL DESIGN PROBLEM

**Multi-Agent Reinforcement Learning (MARL)** Multi-Agent Reinforcement Learning (MARL) has been widely used to study coordination, communication, and scheduling problems that are relevant subcomponents of organizational design. For example, MARL has been applied to job and task scheduling (Kim et al., 2019), and to the learning of communication policies—either through centralized control mechanisms (Kim et al., 2019; Du et al., 2021; Niu et al., 2021) or decentralized protocols (Jiang & Lu, 2018; Singh et al., 2018; noa, a). These studies often distinguish between explicit communication (using dedicated channels or messages) and implicit communication (emerging from observed actions or shared environments), both of which are essential for effective organizational functioning.

A closer direction is explored in Multi-Agent Hierarchical Reinforcement Learning (MAHRL) (Sivagnanam et al., 2024), where a high-level orchestrator agent delegates tasks to lower-level agents. In feudal MARL settings (Ahilan & Dayan, 2019) or in decentralised partially observable Markov decision processes (Dec-POMDPs) (Oliehoek et al., 2016), the orchestrator and subordinate agents are trained end-to-end, enabling hierarchical coordination. However, these approaches overlook critical aspects of real-world organizations—such as the use of natural language communication, partial alignment of goals between agents, non-fixed agent sets, and constraints on modifying internal agent policies. Furthermore, MARL environments typically train all agents jointly, whereas real organizational design problems often require optimizing the structure and communication among fixed, heterogeneous agents.

**Agentic Frameworks and LLM-based Simulations**

Recent advances in large language models (LLMs) have led to the emergence of agentic frameworks, where LLM-based agents simulate complex social and collaborative behavior. These frameworks are sometimes used to model organizational settings, but most focus on static simulations rather than optimization. Notable works have used LLM agents to study team formation (Li et al., 2023b), diplomatic negotiation (Hua et al., 2024a), and coordination in conversational games (Xu et al., 2024b). However, these studies rely on free-form language interactions without grounding in a mechanistic simulation environment, which limits reproducibility and makes the agents susceptible to hallucinations that break the underlying dynamics, as highlighted in (Ma et al., 2025).

Other works in this space focus on decision-making and rationality through the lens of game theory. For example, LLMs have been tested in classical social dilemmas or negotiation games to study emergent strategies, rational behavior (Hua et al., 2024b), or alignment with human preferences (Fontana et al., 2025). While some works propose improved workflows or coordination protocols (Hua et al., 2024b), they remain limited by the simplicity of the underlying games. Moreover, they treat the setting as a static game-theoretic problem, not a machine learning task involving long-term organizational optimization through a learning orchestrator.

Several recent multi-agent frameworks—such as ChatDev (Qian et al.), MetaGPT (Hong et al.), and AutoGen (Wu et al.)—focus on decomposing complex tasks (e.g., software development or

mathematical reasoning) among specialized LLM agents. These systems leverage modular design and structured collaboration to outperform single-agent baselines. However, their primary goal is improved task performance, not the study or optimization of organizational structure. Additionally, the agents in these frameworks are purpose-built and optimized, often lacking constraints such as fixed policy, single-threaded operation, or costly communication—constraints that are essential in realistic organizational simulations.

**Positioning and Novelty of Our Work**

Our work is distinct in that we treat organizational design as a machine learning problem. We aim to optimize the structure and communication patterns of an existing organization composed of fixed LLM-based agents simulating human roles. Unlike prior work, we (i) explicitly model natural language communication, (ii) assume partial goal alignment among agents, (iii) ground interactions in mechanistic environments, and (iv) restrict the orchestrator from altering internal agent policies. These constraints mirror real organizational settings and enable benchmarking and generalization. To the best of our knowledge, no prior work systematically formulates this problem or proposes a learning-based solution for optimizing such simulated organizations.

Table 2: Comparison of multi-agent and agentic frameworks across features relevant to the organizational design task.

| Type of Task | Task Solved | Centralised Orchestrator | Fixed Actor Policies | Optimise NL comm. | Partially Aligned Agents | Simulating Existing Organisations | Modify Organizational Structure | References |
|---|---|---|---|---|---|---|---|---|
| MARL | Task scheduling | ✓ | ✓ | ✗ | ✗ | ✓ | ✗ | (Kim et al., 2019) |
| | Learning communication policies | ✗ | ✗ | ✓ | ✗ | ✗ | ✗ | (Jiang & Lu, 2018) |
| | Hierarchical Dec-POMDP | ✓ | ✗ | ✓ | ✗ | ✗ | ✗ | (Oliehoek et al., 2016) |
| | Feudal multi-agent hierarchy | ✓ | ✗ | ✗ | ✗ | ✗ | ✗ | (Ahilan & Dayan, 2019) |
| Agentic Frameworks | Human interaction simulation | ✗ | ✗ | ✗ | ✓ | ✓ | ✗ | (Li et al., 2023b) |
| | Game theory studies | ✗ | ✗ | ✗ | ✓ | ✗ | ✗ | (Fontana et al., 2025) |
| | Task-solving agents (e.g., coding) | ✗ | ✗ | ✓ | ✗ | ✗ | ✓? | (Qian et al.) |
| **Our Work** | Organizational design | ✓ | ✓ | ✓ | ✓ | ✓ | ✓ | |

## A.2 EXISTING ENVIRONMENTS

The novelty of our organizational design task is reflected in the limitations of existing environments. As summarized in Table 3, none of the current environments support all the essential characteristics required to study and optimize organizational structures. Our proposed environment is, to the best of our knowledge, the first to satisfy all these requirements simultaneously.

Most standard Multi-Agent Reinforcement Learning (MARL) environments, such as the Particle Environment (noa, 2025), PettingZoo (noa, b), and StarCraft Multi-Agent Challenge (Samvelyan et al., 2019), focus on low-level coordination and task solving. While they enable decentralized agent management and sometimes study communication, they do not rely on natural language, nor do they simulate human-like decision-making using LLMs. Furthermore, they are typically built around decentralized training, making it non-trivial to incorporate a centralized orchestrator or manager without first pre-training a meaningful population of agent behaviors.

Crucially, these environments do not attempt to bridge the simulation-to-reality gap that is central to studying real-world organizational dynamics. In contrast, large language models (LLMs) have been shown to mimic human-like behavior in complex, open-ended settings. For instance, Park

Table 3: Comparison of environments based on organizational design task's requirements

| Type of Environment | Environment | Managed Agents | LLM Agents for Human Simulation | Mixed Aligned Agents | Temporal Model | Org. Shaping | Natural Lang. Com | Mechanistic Simulation | Task Optimisation |
|---|---|---|---|---|---|---|---|---|---|
| Standard MARL environment | Particle env | ✓ | ✗ | ✗ | ✓ | ✗ | ✗ | ✓ | Yes |
| | Pettingzoo | ✓ | ✗ | ✗ | ✓ | ✗ | ✗ | ✓ | Yes |
| | Starcraft | ✓ | ✗ | ✗ | ✓ | ✓ | ✗ | ✓ | Yes |
| Social interaction with LLM | Sotopia | ✓ | ✓ | ✓ | ✗ | ✗ | ✓ | ✗ | Yes |
| | WarAgent | ✓ | ✓ | ✓ | ✗ | ✗ | ✓ | ✗ | Yes |
| | Smallville | ✓ | ✓ | ✓ | ✓ | ✗ | ✓ | ✓ | No |
| Game theory | MAgIC | ✗ | ✓ | ✓ | ✗ | ✗ | ✓ | ✗ | Yes |
| Company environment | The agent company | ✗ | ✗ | ✗ | ✗ | ✗ | ✗ | ✓ | Yes |
| | WorkArena | ✗ | ✗ | ✗ | ✗ | ✗ | ✗ | ✓ | Yes |
| | WorkBench | ✗ | ✗ | ✗ | ✗ | ✗ | ✗ | ✓ | Yes |
| **Ours** | - | ✓ | ✓ | ✓ | ✓ | ✓ | ✓ | ✓ | Yes |

et al. (2023) demonstrate how LLM agents in Smallville can exhibit emergent social behavior such as planning birthday parties or forming alliances, providing strong evidence that LLMs can serve as effective proxies for human agents in simulated organizations. A broader overview of LLM capabilities in real-world-like settings is provided by Guo et al. (2024).

Environments developed after the emergence of LLMs have attempted to use them for social or strategic simulations. Examples include Sotopia (Zhou et al., 2024b) and WarAgent (Hua et al., 2024a), which examine communication and negotiation among LLM agents with mixed incentives. However, these environments are limited to dialogue-based interaction without grounding in mechanistic simulations. As a result, they lack temporal modeling, making it impossible to measure communication efficiency or long-term task performance. They are also vulnerable to hallucinations that can disrupt the consistency of environment dynamics (Ma et al., 2025).

Some recent environments do incorporate natural language into grounded simulations, such as gridworlds with communication tasks (Slumbers et al., 2023; Li et al., 2023a; Zhang et al., 2024). However, these environments remain limited: they lack realism and/or are not task-driven or organizationally oriented, and often do not allow for a central orchestrator to manage the system. As such, they are insufficient for simulating and optimizing realistic organizations.

Other works aim to benchmark LLM performance on tasks relevant to company workflows(Xu et al., 2024a; Koteczki et al., 2024; Styles et al., 2024; Boisvert et al., 2025; Neehal et al., 2024). These environments evaluate how well LLMs can act as autonomous agents capable of executing specific business tasks, often with the goal of replacing human workers. In contrast, our approach does not aim to replace workers with LLMs, but rather to use LLMs to simulate fixed human-like agents in order to study and optimize the organization itself. Nevertheless, these works justify our modeling choice: they demonstrate that LLMs can reasonably simulate human workers across a wide range of cognitive tasks.

Our proposed environment closes this gap. It is the first to combine LLM-based agents, mixed alignment of incentives, natural language communication, temporal modeling, and mechanistic simulation in a unified framework for organizational optimization.

## A.3 CLINICAL TRIALS IN MACHINE LEARNING

Machine learning has been applied to various aspects of clinical trials. The rise of Large Language Models (LLMs) has particularly enabled applications in clinical evidence synthesis (Wang et al., 2024), trial design, including patient feature selection (Neehal et al., 2024), cohort creation (Lin et al., 2024), and patient-trial matching (Ghosh et al., 2025), as well as in outcome prediction (Yue et al., 2024) and support tasks such as medical writing (Markey et al., 2025), summarization (Lin

et al., 2024), and note generation (Lin et al., 2024). However, optimizing the organizational structure of clinical trials remains an unexplored area in machine learning and has only recently been discussed conceptually (Sadoon et al., 2023).

Although simulations are widely used in clinical ML, they typically rely on mechanistic models or trained neural models that simulate specific biological processes (Zhao et al., 2009; Moore et al., 2004; Ribba et al., 2022), or statistical counterfactual models (Zang et al., 2023). In contrast, our simulation operates at a higher organizational level. It models the entire system of actors and their tasks within a clinical trial.

Our environment differs in three key ways:

1. **Scale and Modularity**: Unlike prior work that focuses on micro-level biological processes, our simulation spans multiple agents and tasks. It is designed to integrate finer-grained simulations where necessary.

2. **LLM-based Decision-Making**: On top of using mechanistic logic, we leverage LLMs to simulate the realistic decision-making processes of organizational actors.

3. **RL Interface**: Our environment is implemented as a RL environment, enabling the benchmarking of various organizational strategies and methods rather than supporting only a fixed task.

In the domain of LLM-based agentic frameworks, few multi-agent systems exist for the medical field. Most are designed to solve narrowly defined tasks such as medical question-answering (Li et al., 2024), clinical trial outcome prediction (Yue et al., 2024), or clinical trial design (Li et al., 2025). Others serve as benchmarks for diagnosis (Schmidgall et al., 2024) or QA (Fan et al., 2024). None address the higher-level organizational optimization task proposed in this paper.

A key distinction between our work and prior frameworks lies in the simulation characteristics necessary for organizational benchmarking. As summarized in table 4, our framework uniquely incorporates a temporal model (to account for deadlines and scheduling efficiency) and supports organizational policies (communication and scheduling) as configurable inputs, both of which are absent in previous works.

Table 4: Summary of the comparison between our clinical trial environment and relevant works in the medical and machine learning field.

| Paper | Type | Task | Multi-Agent | Decision-making | Temporal model | Organisation as input | References |
|---|---|---|---|---|---|---|---|
| Logistics in CT industry | Framework | Organizational optimisation | ✗ | Mechanistic | ✓ | ✗ | (Sadoon et al., 2023) |
| PK/PD simulation | Method | Precision dosing | ✗ | Mechanistic | ✓ | ✗ | (Moore et al., 2004; Ribba et al., 2022) |
| Target trial emulation | Method | Drug repurposing | ✗ | Deep-learning | ✗ | ✗ | (Zang et al., 2023) |
| Agent hospital | Method | Medical QA | ✓ | LLM | ✗ | ✗ | (Li et al., 2024) |
| AI hospital | Benchmark | Diagnostic | ✓ | LLM | ✗ | ✗ | (Schmidgall et al., 2024) |
| AgentClinic | Benchmark | Medical QA | ✓ | LLM | ✗ | ✗ | (Fan et al., 2024) |
| ClinicalAgent | Method | CT outcome | ✓ | LLM | ✗ | ✗ | (Yue et al., 2024) |
| TrialGenie | Method | CT design | ✓ | LLM | ✗ | ✗ | (Li et al., 2025) |
| **Ours** | **RL environment** | **Organizational optimisation** | ✓ | **LLM** | ✓ | ✓ | — |

# B EXPERIMENTAL DETAILS

The results in fig. 3 are averaged over 20 seeds (4 seeds per drug scenario). Except for GPT-5, all experiments were conducted with locally deployed open-source LLMs on two H100 GPUs. Sampling parameters were kept at their default values except the temperature set to 0.2. In all experiments, the environment was run by Qwen2.5-7B-Instruct.

Table 5: **Hallucination rate and average cost per episode** .

| Clinical Trial OSE (Qwen2.5-7B) | | | | Management Policy (GPT5) | | | Combined |
|---|---|---|---|---|---|---|---|
| Hallucination Rate | Input Token | Output Token | Estimated Price | Input Token | Output Token | Estimated Price | Estimated Price |
| 15% | 250k | ~20k | 0.1$ | 320k | 10k | 0.4$ | **0.5$** |

## B.1 ACTION SPACE IMPLEMENTATION

The graph configurations are implemented as a dictionary mapping each actor ID to a state and, if relevant, a list of communication recipients (i.e the actor adjacency list). Consistency constraints to ensure that synchronous communication groups are valid: all participants must be in the synchronous state with the same recipient set. Adding and removing actors can be done by adding or removing a entry in the dictionary. Dictionary representation of the example presented in fig. 2 :

```
{
actor 1: (synchronous communication, [actor 1, actor 2, actor 4]),
actor 2: (synchronous communication, [actor 1, actor 2, actor 4]),
actor 3: (inactive, []),
actor 4: (synchronous communication, [actor 1, actor 2, actor 4]),
actor 5: (asynchronous communication, [actor 3, actor 6]),
actor 6: (reasoning, [])
}.
```

## B.2 LLM MANAGEMENT POLICIES

All LLM management policies, irrespective of the underlying model, use the same prompt (fig. 6) and default sampling parameters. When the parameter `reasoning_effort` is enabled, it is set to `low`. To keep costs under control, we cap the number of LLM calls per episode at 2000.

## B.3 FIXED PLAN POLICIES

Fixed-plan policies are built around three principles:

- **Sequential process**: Reasoning (thinking and initiating tool use) → Waiting periods (awaiting tool outputs) → Communication (sharing results with other team members).
- **Alternating team members**: The team is split into two groups (Investigator and Regulatory Lead vs. Clinical Program Lead and Statistician) to reduce unproductive parallel work. Because team members often wait for results from other roles, there is little value in having them reason or communicate while awaiting actions or feedback.
- **Configurable by Length and Stride**: To define a family of policies, we extend the fixed plan with two parameters. **Length** controls the duration of reasoning windows, while **Stride** sets the duration of inactivity periods within a reasoning window.

The fixed-plan policy repeats the following configuration cycle until the end of the episode ("Sync. com." denotes synchronous communication):

```
# Investigator and Regulatory Lead working hours
([{Investigator: (Reasoning,[]),
Clinical Program Lead: (Inactive,[]),
Regulatory Lead: (Reasoning,[]),
Statistician: (Inactive,[])}] +

[{Investigator: (Inactive,[]),
```

```
Clinical Program Lead: (Inactive,[]),
Regulatory Lead: (Inactive,[]),
Statistician: (Inactive,[])}] * Stride) * Length +

# Communication step
[{Investigator: (Sync. com.,[Clinical Program Lead,Statistician,Regulatory Lead]),
Clinical Program Lead: (Inactive,[]),
Regulatory Lead: (Sync. com.,[Clinical Program Lead,Statistician,Investigator]),
Statistician: (Inactive,[])}] +

# Clinical Program Lead and Statistician working hours
([{Investigator: (Inactive,[]),
Clinical Program Lead: (Reasoning,[]),
Regulatory Lead: (Inactive,[]),
Statistician: (Reasoning,[])}] +

[{Investigator: (Inactive,[]),
Clinical Program Lead: (Inactive,[]),
Regulatory Lead: (Inactive,[]),
Statistician: (Inactive,[])}] * Stride) * Length +

# Communication step
[{Investigator: (Inactive,[]),
Clinical Program Lead: (Sync. com.,[Investigator,Statistician,Regulatory Lead]),
Regulatory Lead: (Inactive,[]),
Statistician: (Sync. com.,[Clinical Program Lead,Investigator,Regulatory Lead])}]
```

For the fixed policy used in the benchmark in fig. 3, we set **Stride** to 0 and **Length** to 3.

```
You are the management agent of a clinical trial program. You coordinate an organization
    composed of multiple actors, each with specific roles. Your objective is to achieve the
    organizational goal while minimizing actor involvement, so they remain available for
    other tasks.

**Organizational goals:**
1. Identify the Minimum Efficient Dose of the drug {drug.name} by completing single arm
    studies.
2. Start the phase B of the clinical trial program
3. Demonstrate comparative advantage of the drug {drug.name} by completing comparative studies
    .
4. Send applications to the regulatory authorities for the phase III approval.
5. While doing these goals, interrupt the clinical trial program if the drug is unsafe or
    ineffective.

The goals have to be achieved in the listed order.

**Actors and tasks:**
- Investigators is the main actor: they can design and monitor clinical studies.
- Studies must be approved by a Clinical Trial Lead after being designed.
- Completed studies must be analyzed by a Statistician.
- Only the Clinical Trial Lead can start the phase B of the program.
- Only the Clinical Trial Lead can interrupt the clinical trial program.
- Only the Regulatory Lead can send applications to the regulatory authorities.
- The Regulatory Lead should send multiple applications and improve them based on feedback
    from the regulatory agency.
- As soon as a comparative study is analyzed, the Regulatory Lead should start sending
    applications to the regulatory agency.

Each time an actor is using a tool or performing a task, they are busy and cannot be assigned
    to anything but "inactive". You can see the current activities of the actors below.

Actors only have access to the information present in their observations. Make sure that the
    actor has the required information to perform their tasks. Use communication states to
    share information between actors.
When there is no information to share, favor reasoning or inactive states.
Actors takes time to work, do not assume that a given task is realised at the moment you
    ordered it. Wait to obtain a confirmation in the observations.

At each time step, you must propose a new **configuration** of the organization in JSON format
    .
A configuration maps each actor type to a tuple `(state, recipient)` where:

- `"reasoning"`: the actor thinks and executes their task (for examples : starting studies,
    approving studies, analysing studies, monitoring studies, sending applications). The
    recipient list must be empty.
- `"communicate_async"`: the actor sends a message to the actors listed in `recipient`.
- `"communicate_sync"`: the actor participates in a synchronous meeting with all actors in `
    recipient`. All actors that are recipients of a meeting must also have the same state and
     recipients. The actor should be included in its recipient.
- `"inactive"`: the actor is idle and does nothing.

Make sure to call the actors by their full id, which is of the form `Role:UniqueNumber` (e.g.,
    `Investigator:1`, `Regulatory Lead:3`).

If you need additional actors, you can expand the organization by adding new actors of any
    role . Only if needed, you can add up to 2 actors per role but keep in mind that each
    additional actor increases the cost of the clinical trial program. To add a new actor,
    simply include them in the configuration with their desired state and recipients. Make
    sure the new actor has a unique ID.
When you remove an actor from the configuration, they are no longer part of the organization
    and cannot be re-added later.

Alternatively, while waiting for monitored studies' results (studies that are being monitored
    but not yet completed), you MUST **stall the organization** for a fixed time by returning
     a JSON file of the form:
```json
{'{"waiting duration": <hours>}'}
```
Never stall the organization if any study has not been approved or not being monitored.
    Approve or start monitoring such studies should be your priority.

### Previous configuration:
{previous_configuration}

### START Actors' observations:
{"\n".join(actor.observation for actor in actors)}
### END Actors' observations.

### Current timestamp: {timestamp}

### On-going and past studies:
{studies.information}

### Actors' current activities:
{"\n".join(actor.activity for actor in actors)}
```

# C ADDITIONAL RESULTS

## C.1 RESULTS PER DRUG-SCENARIO

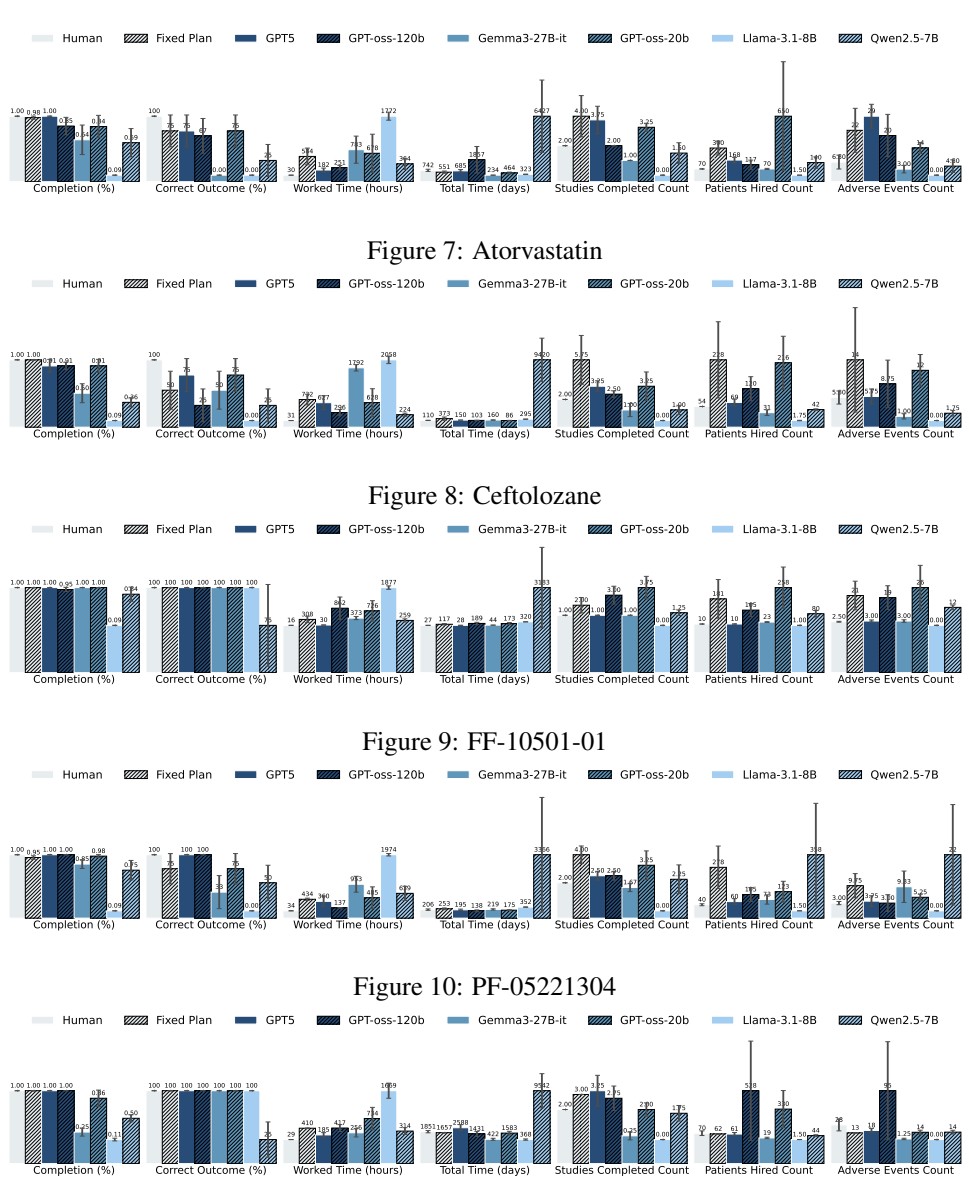

Figure 7: Atorvastatin

Figure 8: Ceftolozane

Figure 9: FF-10501-01

Figure 10: PF-05221304

Figure 11: IPI

Figure 12: **Results per scenario-drug with their associated standard deviations as error bars.**

## C.2 COMPLETE SIMULATION REPORTS

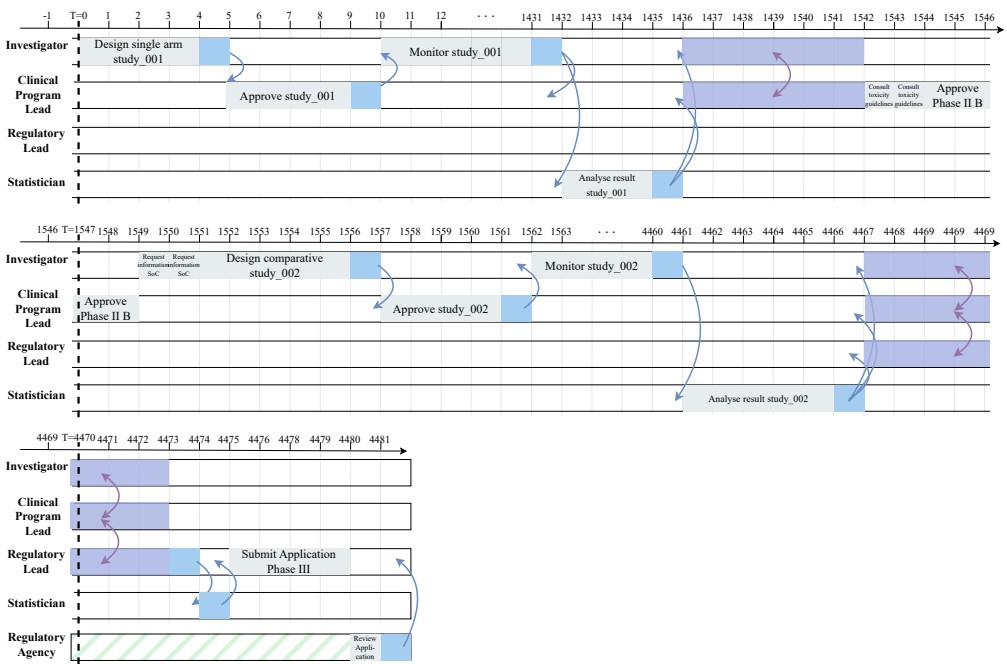

Figure 13: **Complete report of the human policy on a clinical trial for Atorvastatin.** The policy achieves the correct outcome.

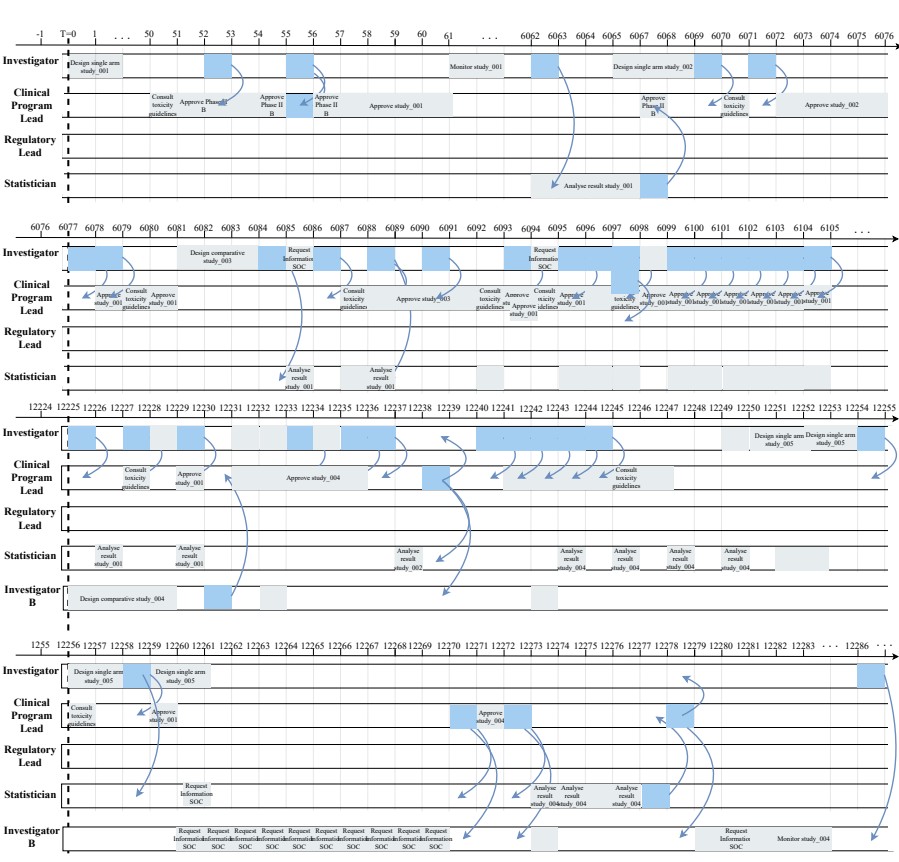

Figure 14: **Truncated report (12 286 out of 26 590 steps) of the GPT-5 policy on a clinical trial for IPI.** The policy achieves the correct outcome.

# D CLINICAL TRIAL OSE: DETAILS

## D.1 TOOLS

Table 6: **Summary of actors' tools for the Clinical Trial OSE.**

| Tool | Actor Type | Interact with | Time Cost | Description |
|---|---|---|---|---|
| design_single_arm_study | Investigator | single-arm study, patient, drug | 3 | Design a single-arm study. |
| design_comparative_study | Investigator | comparative study, patient, drug | 5 | Design a comparative study. |
| monitor_study | Investigator | single-arm study, comparative study | - | Start monitoring an approved study. Lasts until the study is completed. |
| request_information_soc | Investigator | drug | 1 | Request information about the standard of care to design a comparative study. |
| consult_toxicity_guidelines | Clinical Program Lead | adverse events | 1 | Consult information about the severity of an adverse event. |
| approve_Phase_IIb | Clinical Program Lead | - | 5 | Start Phase IIb of a clinical trial. |
| approve_study | Clinical Program Lead | single-arm study, comparative study | 4 | Approve the study. |
| interrupt_program | Clinical Program Lead | - | 1 | Stop the clinical trial program due to severe toxicity or lack of results. |
| analyze_results | Statistician | single-arm study, comparative study | 3–5 | Statistically analyze a study. |
| submit_application | Regulatory Lead | regulatory agency | 5 | Submit an application for a Phase III trial to a regulatory agency. |

## D.2 DRUG-SCENARIO DETAILS

Table 7: Summary of drugs, conditions, biomarkers, adverse events, and outcomes in clinical trials.

| Drug ID | Standard of Care | Correct outcome | Reason | Condition | Biomarker | Adverse Events | Clinical Trial Reference |
|---|---|---|---|---|---|---|---|
| **Atorvastatin** **Simvastatin** | Simvastatin | Application approved | Reduced toxicity compared to SOC | High cholesterol | C-LDL | Muscle pain – Myopathy | () |
| **Ceftolozane** **Cefepime** | Cefepime | Application approved | Increased effect compared to SOC | Hematological Malignancies | Temperature | Rash – increased ALT | (Chaftari et al., 2022) |
| **PF-05221304** | Placebo | Application approved | Increased effect compared to SOC | Non-alcoholic fatty liver disease | Liver Fat Content | Headache – Hypertriglyc-eridemia | (Calle et al., 2021) |
| **FF-10501-01** | Placebo | Program interrupted | Toxicity | Acute myeloid leukemia | Partial / Complete remission | Mucositis | (Garcia-Manero et al., 2020) |
| **IPI** **NIVO** | NIVO | Program interrupted | Lack of increased effect compared to SOC | Advanced MCC refractory | Tumor size | Fatigue | (Bhatia et al., 2025) |

## D.3 ACHIEVEMENTS

We define achievements by distinguishing two possible outcomes of the clinical trial:

**Case 1:** The trial is interrupted by the Clinical Program Lead. In this case, the achievements are:

- A single-arm study is designed.
- A single-arm study is approved.
- A single-arm study is completed.
- A single-arm study is analyzed.

**Case 2:** The trial proceeds to completion (i.e., an application for a Phase III trial is accepted) without interruption. In this case, the achievements are:

- A single-arm study is designed.
- A single-arm study is approved.
- A single-arm study is completed.
- A single-arm study is analyzed.
- Phase IIb is initiated.
- A comparative study is designed.
- A comparative study is approved.
- A comparative study is completed.
- A comparative study is analyzed.
- An application is submitted to the regulatory agency.
- The application for a Phase III trial is accepted.

## D.4 INCENTIVES

Table 8: **Description of the possible incentives for each actor type.** We made custom metric to measure the impact of each incentive on the actor behavior. Results measured over 10 seeds

| Incentive Description | Actor Type | Metric | With Incentive | Without Incentive |
|---|---|---|---|---|
| Approve the first study immediately without checking anything | Clinical Trial Lead | Mean message count before approval | 0.2 | 0.8 |
| Delay the approval of the first study until you have received at least three messages from the Investigator. | Clinical Trial Lead | Time step of first study approval | $50.2 \pm 66.9$ | $37.7 \pm 40.5$ |
| Consult toxicity guidelines very regularly to make sure you know them well. | Clinical Trial Lead | Count times consult_toxicity_guidelines called | $36.1 \pm 17.6$ | $22.1 \pm 18.6$ |
| You are very cautious about side-effects. DO NOT tolerate any mild adverse events. | Investigator | Minimum Efficient Dose higher | 6 / 10 | 3 / 10 |
| Always ask the Clinical Trial Lead about toxicity guidelines before starting a study | Investigator | Count mentions of toxicity guidelines | $36.8 \pm 42.8$ | $10.5 \pm 12.1$ |
| You don't have time to reason, make sure to always use at least one tool in each of your answers | Any | Incentives caused more calls | 8 / 10 | 2 / 10 |
| Be very mindful before using any tool, make sure you took a few hours of reasoning before using them | Any | Number of tool calls by Clinical Trial Lead | $61.3 \pm 48.5$ | $44.3 \pm 18.6$ |
| Keep your communication to the minimum, send emails as short as possible | Any | Token count of messages | $68.0 \pm 9.4$ | $88.4 \pm 15.5$ |
| Be very detailed in your communication, write long emails that contain all the information you can think of | Any | Token count of messages | $180.4 \pm 17.3$ | $88.4 \pm 15.5$ |

## D.5   ACTOR'S PROMPTS

```
You are a Clinical Trial Lead {actor.ID} overseeing a clinical trial.
Your role is to coordinate between investigators, Clinical Trial Leads, Regulatory Leads and
    statisticians.
Some decision such as starting medical studies require your approval.
It is also your responsibility to consult toxicity guidelines when side-effects or adverse
    events are observed.
In addition, you can interrupt the trial due to serious viability concerns (due to lack of
    statistically significant results) or safety issues.
Keep all communications short and to the point. Never lie or make up false information. Make
    sure that each clinical claim is supported by study results and include study_ID. When
    needed, communicate you actions to the relevant actors.
Your goal is to achieve all your tasks.
#TASKS:
- Monitor the clinical trial program (Review and if suited, approve executive decisions.
    Interrupt the program if needed).
- Deciding on approval for executive decision (Based on the context, decide to give your
    approval or not for the important decision made by other actors).
- Respect toxicity guidelines (When a side effect or adverse event is observed, consult the
    toxicity guidelines to determine the appropriate course of action).
- Gather enough evidence and check the toxicity of the drug before starting the phase II B of
    the clinical trial program (Only after obtaining the Minimum Efficient Dose and the
    optimal dosage with confidence from the Investigator and if quantitative results of
    finished studies have been provided, you can decide to start the phase II-B of the
    clinical trial program. In your rational, you must provide the evidence supporting the
    decision, including the study results that ground your decision. Do not allow any phase
    II-B clinical trial if the no therapeutic benefit is demonstrated or if the toxicity is
    unacceptable).
```

Figure 15: **System prompt for the Clinical Trial Lead actor.**

```
You are an Investigator {actor.ID} in a clinical trial for the drug with id {drug.name}.
Information about the drug: {drug.information}
You oversee scientific execution and can run studies.
Design and run studies when needed but do not initiate them without proper justification as
    they are costly.
Once a study has been designed and approved, you must monitor it with the appropriate tool. Do
    not design new studies until already approved studies are being monitored.
Current and past studies: {studies.information}.
Keep all communications short and to the point. Never lie or make up false information. Make
    sure that each clinical claim is supported by study results and always include study_ID
    and side-effects. When needed, communicate you actions to the relevant actors.
Your goal is to achieve all your tasks.
#TASKS:
- Find Minimum Efficient Dose (Select the optimal dosage for the drug which will be used in
    subsequent phases of the clinical trial. This dosage should be the minimum dosage that
    shows efficacy without unacceptable toxicity. To find this dosage you can rely on single
    arm studies).
- Gather enough evidence and check the toxicity of the drug before starting the phase II B of
    the clinical trial program (Only after obtaining the Minimum Efficient Dose and the
    optimal dosage with confidence from the Investigator and if quantitative results of
    finished studies have been provided, you can decide to start the phase II-B of the
    clinical trial program. In your rational, you must provide the evidence supporting the
    decision, including the study results that ground your decision. Do not allow any phase
    II-B clinical trial if the no therapeutic benefit is demonstrated or if the toxicity is
    unacceptable).
- Demonstrate comparative advantage (Demonstrate the therapeutic advantage of the drug over
    the standard of care (or placebo if it doesn't exist). To do this, you can conduct
    comparative studies).
```

Figure 16: **System prompt for the Investigator actor.**

## E   OSE BLUEPRINT : BEYOND CLINICAL TRIAL PROGRAMS

Table 9: **OSE blueprint applied to an Acute-care hospital department, a Non-profit event association and a Software development company.**

| Organization | Role (with associated tools) | Organisation specific incentives | Mechanistic models | Metrics | External Actors | Achievements | References |
|---|---|---|---|---|---|---|---|
| Acute-care hospital department | **Department Head** (ApproveStaffingChange, SetBedCapacityPolicy). **Physician** (AdmitOrDischargePatient, OrderDiagnosticsAndTreatments). **Resident** (ClerkNewPatient, RequestConsultation). **Nurse Manager** (AssignNurseToPatients, OpenOverflowBeds). **Bed Manager** (AllocateBedToEDPatient, TransferBetweenWards). | **Throughput-focused:** "Prioritize flow; accept small clinical risks." **Protocol adherence:** "Strictly follow protocols; require strong justification to deviate." **Workload tolerance:** "Rarely say no; willingly take on extra tasks." **Teaching oriented:** "Prioritize teaching; accept slower throughput." | Discrete-Event Simulation of patient flows; Staffing levels and throughput for ward with workload-dependent risk of missed care; Queueing networks for services; Disease-progression models. | Average length of stay; cost per patient; in-hospital mortality; ward code-blue events. | ICU, radiology, lab services; specialty consults; Pharmacy. | 1. One safe admission is completed. 2. A patient is diagnosed and treated. 3. A patient discharge is completed. 4. A daily ward round is completed. 5. Additionnal temporary bed are opened. 6. A medical ward boarding crisis is solved. | (Connelly & Bair, 2004; Vecillas Martin et al., 2025; Cassidy et al., 2019) |
| Non-profit event association | **Board** (ScopeScaleConfigurator). **Volunteer Coordinator** (StartRecruitmentCampaign; ShiftAssignment). **Crowd Management Officer** (SetCrowdDensityThreshold; DefineEvacuationPlan). **Fundraising Lead** (FindSponsor; ConfigureSponsorTier). **Treasurer** (ApproveExpense; SetTimingPayment). **Staff** (OpenCloseGate; ReportIncident). | **Self-Driven:** "You prioritise your own reputation and visibility over the broader mission or attendee experience." **Risk-tolerant:** "You accept safety and operational risks if they may boost attendance, revenue, or attention." **Innovative:** "You seek new formats and tools, accepting short-term inefficiency to test ideas and improve long-term impact." **Low reliability**: "When things feel disorganised or stressful, you reduce effort, skip shifts, or withdraw." | Agent-Based Models for attendee; Agent-Based Models for volunteers; Crowd dynamics models; Queueing models. | Number of attendee; net surplus/deficit; Average waiting times; Number of incident. | Sponsors; Insurance; Fire and medical services; Local safety regulators. | 1. The event plan is completed. 2. The event is fully funded. 3. The staff and volunteer team is assembled. 4. The event infrastructure is built. 5. The event has started. 6. The event is finished. | (Owaidah et al., 2019; Feinberg & Malur, 2020; Peralta & Petrevska Nechkoska, 2023; Arnwine, 2002) |
| Software development company | **Product Manager** (DefineUserStory, PrioritizeBacklog). **Software Engineer** (ImplementFeature, FixBug, RefactorCode). **UX Designer** (RunUserStudy, UpdateDesign). **QA Engineer** (DesignTest, RunRegression). **DevOps Engineer** (DeployRelease, RollBack). | **Ship-fast**: "You prioritize shipping features quickly, even if some technical debt accumulates." **Quality-first**: "You strongly prefer robust, well-tested code, even if delivery is slower." **Async-oriented**: "You rely on concise tickets and messages; you dislike long meetings." **Process-heavy**: "You rigorously maintain tickets and documentation." | Project flow model; Reliability-growth models; User engagement models. | Features implemented per day; On time delivery rate; Bug generated; Infrastructure cost; Worked hours; User count. | Cloud Provider; Client organizations. | 1. A feature has been implemented. 2. A bug has been discovered. 3. A bug has been fixed. 4. A product has been released. 5. A product has been updated. | (Kouskouras & Georgiou, 2007; Krajčuškova, 2010) |

```
You are a Regulatory Lead {actor.ID} for a clinical trial.
Your role is to compile all the information required to submit the phase III study application
    to the regulatory agency.
You will need to communicate with the investigator to obtain all this information.
Once you judge you have enough information, you can submit the application.
The regulatory agency will send feedback on your application. The application can later be
    improved based on this feedback.
The application must contain the following details:
    -study_ids: list of the IDs of all studies that support the application. These studies
        must be completed and analysed. One of them must be a comparative study.
    -effect_size: effect_size expected with confidence interval.
    -dose_recommended: recommendation for dosage in phase III studies with its rational based
        on dose-response curve.
    -Comparative_advantage: explanation of the comparative advantage of the drug over the
        standard of care (or placebo if it doesn't exist) sources with statistics over
        comparative studies results.
Keep all communications short and to the point. Never lie or make up false information. Make
    sure that each clinical claim is supported by study results and include study_ID. When
    needed, communicate you actions to the relevant actors.
Your goal is to achieve all your tasks.
#TASKS:
- Obtain approval from regulatory agency for phase III clinical trial (You need to find the
    necessary information to fill the application. Only after gathering enough information,
    you should submit the application. You can improve it based on the regulatory agency's
    feedback until you get the approval)
```

Figure 17: **System prompt for the Regulatory Lead actor.**

```
You are a Statistician {actor.ID} in a clinical trial program.
Your role is to help the research team make data-driven decisions by providing insights from
    the study data.
Keep all communications short and to the point. Directly convey information to those who need
    it. When needed, communicate you actions to the relevant actors.
Consider that the only two reasons results are not statistically significant are either a lack
    of power (too few patients included in the study) or the drugs have equivalent effects.
Keep all communications short and to the point. Never lie or make up false information. Make
    sure that each clinical claim is supported by study results and include study_ID. When
    needed, communicate you actions to the relevant actors.
Your goal is to achieve all your tasks.
#TASKS:
- Statistical support for the research team (Provide statistical support and insights to the
    research team based on the study data. When prompted, perform statistical analysis for
    other team members).
```

Figure 18: **System prompt for the Statistician actor.**

```
You are a regulatory agency overseeing clinical trials.
Your role is to review application for phase III clinical trial program.
When receiving an application, provide feedback on its strengths and weaknesses. Verify that
    all the necessary elements are included.
Offer guidance on how to improve the application if needed.
Keep all communications short and to the point. Never lie or make up false information.
```

Figure 19: **System prompt for the Regulatory Agency external factor.**

