# OpenReview forum: "Building Simulation Environments for Computational Organizational Design"
_ICLR.cc/2026/Conference — Submitted to ICLR 2026_

### Official Review · Reviewer_3QUG · 2025-10-28

**Soundness:** 2
**Presentation:** 4
**Contribution:** 3
**Rating:** 4
**Confidence:** 3

**Summary:**

This paper introduces the Organizational Design Problem (ODPs) and attempts to learn a management policy that configures the team composition, communication, and autonomy to achieve multi-objective goals under structural constraints. A key contribution of this work is the introduction of Organizational Simulation Environments (OSEs), which couple domain-specific mechanistic models with LLM agents that communicate in natural language within a discrete-time simulation. The authors work with experts to create a Clinical Trial OSE that models up to 25 different actors and 8 drugs across several scenarios and benchmark baseline management policies. The results present interesting takeaways for future work in solving ODPs.

**Strengths:**

+ The formulation of the Clinical Trial OSE is interesting and well-formulated. This OSE is more complex than typical simulation environments due to the multiple actors and variables.
+ The evaluation is interesting and includes a comparison to a human management policy. The results show that there is still work to be done in addressing complex ODP problems with AI techniques.

**Weaknesses:**

- There is no validation to ensure that a real-world system (and all its components) will transition in the way the LLM specifies.
- It is unclear how repeatable such a simulation will be. If a simulation is run with the same start state, same management policy, and same LLM, will the output be the same?

**Questions:**

- Can you provide evidence that the proposed Clinical Trial OSE behaves similarly to the real-world Clinical Trial with humans?
- Can you provide information regarding the repeatability of the simulation?
- Can you tell me which ODP environments such a framework is well-suited for and which it is not?

---

> ### Author Response · Authors · 2025-11-21
>
> We thank this reviewer for taking the time to review our work. We have carefully considered each point of feedback and provide our point-by-point responses below. Please don’t hesitate to let us know if any further clarifications are required.
>
> **P1 Sim-to-real gap**
>
> The reviewer’s formulation “transition in the way the LLM specifies” is confusing in our setting, because environment transitions in our simulator are governed by mechanistic models, not by LLM-generated content. The only reliance on LLM outputs is for simulating the behaviour of human-like actors (internal or external to the organisation). All other non-human components of the simulator (for example, in the Clinical Trial OSE: disease progression, adverse events, comparative efficacy, etc.) are implemented as mechanistic models taken from the literature and calibrated to our simulator. These deterministic models do not depend on LLM outputs and therefore do not require validation for hallucination and do not have to perform out-of-distribution generalisation as the simulator is built around them. Their structure and parameters have been reviewed and validated by experts from the pharmaceutical industry (whose names will appear in the camera-ready version). Details of the mechanistic models are summarised in Table 1, and the clinical trial references used for calibration (for example, for selecting the shape and parameters of the dose–response curve) are listed in Table 7.
>
> As stated above, LLMs are only used to model human actors. Multiple previous studies have evaluated whether LLM-based agents provide credible human-like behaviour along dimensions such as believability, knowledge, handling of secrets, relationships, social rules, financial and material trade-offs [1], as well as self-knowledge, memory, plans, reactions, and reflections [2]. In both studies, LLM agents obtain scores comparable to human participants on these criteria, leading the authors to conclude that LLM-based actors offer believable behaviour and are reasonable proxies for human judgement and decision-making. Together with the mechanistic grounding and expert validation of the non-human components, this supports our claim that the Clinical Trial OSE captures key qualitative features of real-world Phase~II clinical trial programmes, while remaining an abstraction designed for controlled experimentation on organizational design.
>
> [1] (2023) SOTOPIA: Interactive Evaluation for Social Intelligence in Language Agents
>
> [2] (2023) Generative Agents: Interactive Simulacra of Human Behavior
>
> **P2 Reproducibility concern**
>
> We understand the reviewer’s concern regarding the reproducibility of our environment due to the LLM stochasticity for the actor. In our implementation of the OSE environment, we propose relying either on API calls to cloud providers or locally hosted LLMs (via vllm). The latter enables reliable seeding, which is why our benchmark is created with a locally deployed LLM (namely Qwen-2.5-7B) for the actor’s policy. In that case, if the management policy is also seed-controlled (which is the case for the Fixed Plan policy and all the LLM policies except GPT5), we can confirm that running an episode for a given seed will always output the same result.
>
> **Action taken.** Clarification regarding the reproducibility of the experiment in the clinical trial OSE is added in Appendix B.

---

> ### Author Response · Authors · 2025-11-21
>
> **P3 Which ODP environment is the framework suited for?**
>
> Our framework is, by design, general and can in principle, be applied to a wide range of organizations. However, it is particularly well-suited to ODP environments where the organization operates through relatively fixed, clearly defined processes or sequences of tasks. In such settings, the substantial engineering effort required to design and calibrate mechanistic models for non-human resources and external factors can be amortized over many runs and variations of the same underlying process. By contrast, organizations that frequently and radically change their core processes, or that rely almost entirely on ad-hoc, non-procedural coordination, are less well suited to our current blueprint. In the former case, the diversity of processes would require continuously redesigning a large number of mechanistic models; in the latter, the absence of stable process constraints would lead to an explosion in scenario complexity for a single environment instance.
>
> To make this distinction more concrete and to illustrate which organizations are favourably modelled by OSEs, we have added a new appendix in which we apply the OSE blueprint “on paper” to three additional organizational types: an acute-care hospital department, a non-profit event association, and a software development company. While we do not provide full implementations for these examples, we systematically specify each of the components defined in the OSE blueprint for each organization. This both clarifies the range of ODP environments for which the framework is well-suited and makes its generality more explicit, helping readers see how to adapt the OSE template to their own organizational settings.
>
> **Action taken.** We indicate in the limitation section that our OSE blueprint is particularly suited for organisations with clear processes. And Table 9 in Appendix E presents the application of the OSE blueprint to three new types of organisations. (This table can directly be seen at this [link](https://imgbox.com/bHBDXFyc)).

---

### Official Review · Reviewer_XFAh · 2025-10-31

**Soundness:** 2
**Presentation:** 2
**Contribution:** 2
**Rating:** 2
**Confidence:** 3

**Summary:**

This paper focuses on the formulation of the Organizational Design Problem and the introduction of a corresponding testbed, Organizational Simulation Environments (OSEs). These environments are governed by a mechanistic, discrete, domain-specific logic (e.g., elapsed time). The authors present a complex case study modeling a Clinical Trial and its associated organizational challenges.

**Strengths:**

- Addresses a crucial and challenging optimization problem within management and organizational science.
- Introduces a complex, multi-faceted case study (the Clinical Trial OSE) to benchmark organizational policies.
- Provides a comparative analysis of several pre-trained language models (LLMs) in this complex coordination task.

**Weaknesses:**

1. Mismatched Use Case for Organizational Design: The paper frames the challenges of clinical trials as resulting primarily from management and organizational design, yet it fails to substantiate this premise with supporting literature. This focus is questionable, as it ignores established research (e.g., Sun et al., "Why 90% of clinical drug development fails and how to improve it?")
2. Mismatch Between General Problem and Specific Implementation: There is a disconnect between the ambitious, general definition of the Organizational Design Problem (ODP) and its sole, highly specific implementation (a clinical trial). The paper fails to provide a clear path for adapting the framework to other organizational types, even simpler ones, thus leaving the claims of generalizability unsubstantiated.
3. Absence of Baseline: The task is defined as "learning a management policy," yet the paper demonstrates no learning or optimization process. The experiments are limited to evaluating pre-trained LLMs in a zero-shot, prompted setting. A crucial baseline attempting actual policy optimization (e.g., via RL) is missing.
4. Questionable Benchmark Complexity: The fact that pre-trained models achieve 100% success without any fine-tuning strongly undermines the paper's claims about the benchmark's complexity and difficulty.
5. Unanswered Questions: The introduction poses questions (e.g., "How should teams be structured?", "What communication policies minimize costly delays?"). However, the paper provides no answers or actionable insights toward solving them with OSE.
6. Unverified LLM Reliability: The simulation's success critically depends on the LLMs' ability to reliably simulate actors and use tools. The paper provides no information or analysis verifying this, ignoring known LLM limitations in consistency and tool-use fidelity.

**Questions:**

- The paper states real organizations have heterogeneous individuals (diverse personalities, skills, and experience). Why were these variations in agent attributes not implemented in the simulation?
- What specific, actionable conclusions, if any, can be drawn from this single use-case simulation for the organizational design of real-world institutions?
- The results show variance in performance. What factor appears to be the primary driver of organizational success in the simulation: the level of autonomy, the defined communication policy, or the individual agents' competence in task execution?

---

> ### Author Response · Authors · 2025-11-21
>
> We thank this reviewer for taking the time to review our work. We have carefully considered each point of feedback and provide our point-by-point responses below. Please don’t hesitate to let us know if any further clarifications are required.
>
> **P1 Clinical trials don't fail because of the organization**
>
> We thank the reviewer and agree that the work of Sun et al. (2022) is highly relevant to motivating our environment, and we would like to emphasize that we already compare and engage with it in the original manuscript. As noted in the paper, we explicitly cite and discuss Sun et al. in the motivation section (lines 286 and 293), and several of the figures presented in our manuscript are directly based on this work, making it one of the main references supporting our use case. In particular, Sun et al. report that “poor strategic planning accounts for 10% of drug development failures,” which, given the cost, duration, and public-health importance of clinical trial programs, corresponds to a substantial and practically meaningful source of avoidable failure.
>
> **P2 Mismatch Between General Problem and Specific Implementation**
>
> First, we clarify that the paper explicitly introduces an OSE blueprint that enumerates a set of core components required to instantiate an environment for any organization. This blueprint is designed to provide a structured path from the abstract ODP definition to concrete domains. Section 2.2 explains, for each component, how it can be specified in a domain-agnostic way, and the Clinical Trial OSE in Section 3 is presented as a single, detailed instantiation of this general blueprint rather than a restriction of the framework to clinical trials.
>
> Second, we understand the reviewer’s concern that the second half of the paper, being centered on the clinical trial example, may not convincingly demonstrate the generality of the proposed template. To address this, we have added a new appendix in which we apply the OSE blueprint “on paper” to three additional organizational types: an acute-care hospital department, a non-profit event association, and a software development company. While we do not provide full implementations for these examples, we systematically specify each of the components defined in the OSE blueprint for each organization. This makes the generality of the framework more explicit and is intended to help readers see how to adapt the OSE template to their own organizational settings.
>
> **Action taken.** The application of the OSE blueprint to three new types of organization is presented in Appendix E of the revised manuscript. (This table can directly be found at this [link](https://imgbox.com/bHBDXFyc)).
>
> **P3 Absence of learned baselines**
>
> We would like to clarify the scope of this work. This is a benchmark and dataset paper: our primary contribution is to define the ODP and to introduce an environment that enables the development and evaluation of management policies. At no point do we claim to solve the introduced task with a learned policy. Consistent with this, the current experiments are restricted to evaluating pre-trained LLMs used as management policies in a zero-shot, prompted setting, rather than training new policies.
>
> Regarding the reviewer’s suggestion to include an RL baseline, we emphasize that constructing such a baseline is itself highly non-trivial in our setting. As the reviewer notes, the task is defined over: (i) natural language interaction (which makes the use of a pre-trained LLM mandatory), (ii) very long horizons (over 10k steps per episode), (iii) vectorial reward functions, and (iv) overall organizational complexity (multiple heterogeneous actors, substantial domain knowledge not taught by the environment). To the best of our knowledge, no existing RL method is able to handle all of these constraints jointly. Designing and validating such a learning algorithm would therefore constitute a substantial, separate contribution, which lies beyond the scope of this environment-focused paper.

---

> ### Author Response · Authors · 2025-11-21
>
> **P4 Benchmark complexity**
>
> We believe this concern arises from a confusion about what “success” means in our benchmark. Most RL environments summarize performance with a single scalar metric. In contrast, an OSE is explicitly multi-objective: the organization must jointly optimize several competing goals. This is why we formalize the ODP as a Multi-Objective POMDP, and why no single “success” number captures the difficulty of the task.
>
> In the Clinical Trial OSE, while we are not entirely sure what the reviewer refers to by “100% success,” we assume it relates to GPT-5 achieving a 98% completion rate and/or 90% correct outcomes. As explained in Section 3.3 and Appendix D.3, the completion rate is a sanity-check metric: it only verifies that the management policy is able to finish a clinical trial before the time limit, independent of how well the trial was managed. Because management policies can get stuck in loops or hallucinations and never complete the scenario, we need this metric to ensure that episodes terminate so that any analysis is meaningful. A high completion rate therefore does not indicate that the benchmark is easy; it only indicates that the policy can produce coherent trajectories.
>
> The Correct outcome metric evaluates whether drugs that are too toxic or lack therapeutic benefit (as defined by medical guidelines and the mechanistic models) are correctly stopped or and other “good” drugs advanced to Phase III. Here, anything short of 100% would in practice be unacceptable: by design, the Phase II decision logic in the OSE follows the scientific method (clinical studies and statistical analysis of outcomes), and this logic is not LLM-generated but implemented via mechanistic models. In this context, a high correct-outcome rate is a necessary safety property, not evidence that the environment is trivial.
>
> The substantive difficulty of the benchmark lies in optimizing the remaining dimensions of the vectorial reward under these safety and correctness constraints. To clarify the role of each dimension and highlight the remaining performance gap, we summarize the metrics and the behavior of the GPT-5 management policy relative to the human policy:
>
> - **Worked time:** Directly linked to the cost of running the trial; administrative staff alone account for about 30% of total clinical trial program costs [1]. The GPT-5 policy uses around 10× more worked hours on average than the human policy.
> - **Total time:** Linked to the effective exploitation window of the drug due to patent limits, and thus to the commercial viability of the program [2]. The GPT-5 policy leads to trials that are on average 24% longer.
> - **Patient hired count:** Closely tied to trial cost, since patient recruitment and monitoring account for about 15% of clinical program costs [1]. The GPT-5 policy hires about 50% more patients on average than the human policy.
> - **Adverse event count:** Directly related to ethics and patient well-being, which are central constraints in clinical development. The GPT-5 policy generates approximately 30% more adverse events than the human policy on average.
> - **Studies completed count:** A main driver of patient count, total time, and adverse events. The GPT-5 policy requires about 50% more studies than the human policy on average.
>
> A management policy that achieves high completion and correct outcome but does so with dramatically inflated costs, longer timelines, and more adverse events is not satisfactory for clinical trial management. Our benchmark is designed precisely to expose these trade-offs. Therefore, the fact that a strong pre-trained model comes close to “success” on a narrow safety/efficacy dimension does not undermine the complexity or difficulty of the benchmark. Instead, it shows that even powerful LLMs still struggle to solve the multi-objective organizational design problem encoded in the OSE, where time, cost, and patient well-being must all be optimized simultaneously.
>
> [1] (2016) *Key cost drivers of pharmaceutical clinical trials in the United States*
>
> [2] (2017) *How do patents affect research investments?*

---

> ### Author Response · Authors · 2025-11-21
>
> **P5 Unanswered questions**
>
> First, we would like to clarify that the questions in the Introduction are intended as illustrative and motivational examples to introduce organizational science. The aim of this paper is not to fully answer these broad and longstanding questions, but to provide tools (both a formalism and a class of environments) that are necessary to study such questions with machine learning methods in a controlled and repeatable way.
>
> Second, we understand the reviewer’s disappointment at not seeing these broad questions explicitly resolved, and we agree that the framing in the original Introduction could be better aligned with the scope of our results. We have therefore reformulated and narrowed the illustrative questions in the first paragraph to better match the concrete insights from Section 4. The new questions are: “How should team schedules be structured to achieve different objectives? What is the impact of communication policies on costly delays? How robust is an organization to individual mistakes?” Within this narrower scope, Section 4 does provide partial answers and actionable insights. In particular, (i) by exploring the Pareto frontier of the Fixed Plan family, we show how different meeting and reasoning schedules trade off worked time and total time; (ii) by comparing GPT-5 and human management policies, we highlight how GPT-5’s communication patterns induce additional delays and noisy active states; and (iii) by contrasting how humans and GPT-5 respond to actor errors in adverse-event handling, we show that GPT-5 policies are substantially less robust to individual mistakes. Together, these results demonstrate how the Clinical Trial OSE can be used to generate concrete organizational design insights about scheduling, communication policies, and robustness, rather than only posing abstract questions.
>
> **Action taken.** In the revised paper, we have made the illustrative questions in the Introduction more specific and aligned with our empirical results, and we have updated the wording in Section 4 to explicitly connect the reported findings to these questions.

---

> ### Author Response · Authors · 2025-11-21
>
> **P6 Hallucinations and tool usage**
>
> We would like to clarify that the paper already presents, in Section 4, a discussion of hallucinations in the actor policies, together with a quantitative analysis. We report an overall hallucination rate of 15%, and show that these hallucinations are concentrated in only 10% of the episodes, as a single hallucination often leads to a short cascade within the same episode.  Moreover, in the revised paper, we now provide an additional description of the hallucinations encountered in our study. We classify the trigger of each hallucination (counting only the first hallucination that initiates a cascade) and find that hallucinations are particularly localised around manipulating studies. Specifically, 17% of hallucination triggers are due to hallucinating the wrong study identifier, 33% are due to hallucinating a new study, and 50% are due to hallucinating the results or reviews of a study that exists in the environment but has not yet been run. This analysis clarifies that hallucinations affect how agents reference and reason about specific studies, rather than arbitrarily corrupting all aspects of the simulation.
>
> Regarding tool-usage reliability, we also report statistics on the rate of failed tool calls. As expected given known limitations of LLMs, particularly smaller 7B models, the rate of failed tool calls is relatively high at 47%. However, these failures are mitigated by standard LLM prompting methods: we use a reflection framework in which tool-failure feedback is returned to the LLM, the model reflects on the failure, and this reflection is added to the context to guide subsequent tool calls. The performance of the human policy, which interacts with the same LLM actors and thus experiences the same tool-failure modes as the LLM policies, shows that these limitations do not prevent substantially better performance than what the LLM policies achieve. This indicates that our main conclusions are not driven by systematic tool failures, but by genuine differences in organizational policies.
>
> Finally, while hallucinations and tool-use errors are indeed undesirable side effects of using LLMs, simulating human behaviour with LLM-based agents still represents a substantial improvement over previous approaches. Traditional agents based on hand-crafted rules, finite-state machines, or behaviour trees tend to be not robust, hard to scale, and repetitive, whereas LLM agents leverage broad learned knowledge and natural-language memory to generate flexible, context-appropriate, and coherent long-term behaviour without exhaustive scripting. Game-style RL agents typically optimise explicit rewards and often learn non-human, exploitative strategies, while LLM-based agents can follow informal social goals and norms expressed in text, producing more human-like patterns such as planning events, coordinating, negotiating, and forming relationships without extensive task-specific reward design. Moreover, previous systems generally required special-purpose scripting or reward design to specify goals and policies, whereas LLM-based agents can be guided directly through natural-language descriptions of roles, objectives, constraints, and scenario changes, which greatly simplifies the design and scaling of multi-agent social simulations.
>
> **Action taken.** Statistics on tool-usage failures have been added to Section 4 of the revised manuscript.
>
> **P7 Implementation of heterogeneous individuals**
>
> We agree that having heterogeneous individuals is an important aspect of human-like organizations. We would like to clarify that such heterogeneity is in fact implemented in our Clinical Trial OSE through the incentive mechanism described in sections 2.1, 2.2.3 and 3.1. Concretely, we modulate each actor’s behavior by adding incentive-specific information to the prompt that governs its actions. This yields distinct behavioral profiles for actors in the same role. We demonstrate in an ablation study in Appendix D.4 that modifying these incentive prompts changes the actors’ behavior as expected. The full list of incentives is provided in Table 8; examples include “Keep your communication to the minimum, send emails as short as possible” and “You are very cautious about side-effects. DO NOT tolerate any mild adverse events.”

---

> ### Author Response · Authors · 2025-11-21
>
> **P8 Actionable conclusions**
>
> We agree that a single use-case simulation cannot yield definitive prescriptions for all real-world institutions. However, the Clinical Trial OSE provides several specific, actionable conclusions for organizational design:
>
> First, it delivers concrete artifacts for practitioners and researchers: a new environment and a new framework (the OSE blueprint) that can be instantiated beyond clinical trials. These make it possible to systematically learn, test, and compare management policies.
>
> Second, the benchmark clearly shows that current LLM management policies remain far from human performance in terms of efficiency and safety, even when they can complete the task. This gap indicates that specialized organizational-level methods need to be developed/learned.
>
> Third, the simulation clarifies why such methods are non-trivial to train: (i) organizational control must be expressed in natural language, which makes the use of pre-trained LLMs essentially mandatory; (ii) the decision process is very long-horizon (episodes span >10k steps); (iii) objectives are inherently vector-valued, requiring explicit handling of multi-objective (vectorial) reward functions; and (iv) the system must coordinate multiple heterogeneous actors and resources. Together, these properties highlight concrete technical requirements that any realistic organizational-design method will have to satisfy.
>
> Fourth, our systematic analysis of failure modes of pre-trained LLM policies yields actionable design principles for future methods:
>
> - Management policies require domain-specific field knowledge that is can not be  learned efficiently from the environment alone, suggesting that organizational policies must have access to such knowledge.
> - Management policies must learn when actors should remain inactive, rather than defaulting to constant action, to avoid noisy over-activity that wastes resources and creates downstream errors.
> - The timing and structure of communication between actors are critical: missing or poorly timed messages produce large downstream coordination failures, emphasizing the need to explicitly design and optimize communication policies.
> - Fixed, non-adaptive policies are inefficient: rigid schedules of meetings and reasoning steps fail to align with the stochastic dynamics of the organization, motivating adaptive, state-dependent management policies.
> - Finally, organizations must be designed to be robust to local actor mistakes, with mechanisms that detect and correct errors rather than propagating them through the system.
>
> Although obtained in a single clinical-trial OSE, these conclusions constitute concrete, testable design principles and training requirements that can be directly used to guide the development and evaluation of organizational policies in other real-world institutions.
>
> **Action taken.** In the revised manuscript, we highlight these insights as the main takeaways from our experimental analysis.
>
> **P9 What factors lead the most to success?**
>
> With the current experiments, we cannot identify a single primary driver of organizational success among autonomy, communication policy, and individual agent competence.
>
> First, these dimensions are tightly coupled, leading to substantial confounding: changes in one dimension typically induce changes in the others. For example, specific incentives that shape actors’ communication behavior will change which communication policies are effective, while a given management policy may iteratively adjust team composition until it finds actors whose incentives match its preferred communication pattern. Moreover, except for the Fixed Plan policies, the baselines in our benchmark do not explicitly parameterize autonomy, communication, or competence; for these policies, we only observe the final organizational behavior and outcomes.
>
> Second, each factor influences different components of the multi-objective optimization problem, sometimes in conflicting ways. As illustrated in Figure 5, varying autonomy by giving actors more or less time to reason leads to different optima across metrics such as Worked Time and Total Time, so a priori no single setting dominates across objectives.
>
> Third, episodes in the Clinical Trial OSE are long (over 10k steps) and heterogeneous: different stages of the trial require different interaction patterns, and a given policy may be well adapted to some tasks while underperforming on others. This further complicates attributing overall success to a single factor.
>
> For these reasons, our analysis is restricted to examining specific interaction patterns in the full interaction reports, as exemplified in Figure 4. Only fine-grained analyses that highlight the inadequacies of particular policies are currently possible, and no general conclusion can be drawn that autonomy, communication policy, or individual competence is the single primary driver of organizational success in our simulations.

---

### Official Review · Reviewer_ppce · 2025-11-02

**Soundness:** 1
**Presentation:** 2
**Contribution:** 2
**Rating:** 2
**Confidence:** 4

**Summary:**

The authors introduce a novel research direction for organizational design based on reinforcement learning. They aim to develop a management policy that governs consolidation and communication among multiple actors, and shows enhanced robustness to unpredictable events and varied organizational contexts. The authors’ main contributions are: (1) modeling the organizational design problem as a POMDP; (2) proposing a blueprint for developing environments to study this problem; and (3) providing an instantiation of a clinical trial scenario while also evaluating different LLM-based management policy baselines on this scenario using multiple metrics.

**Strengths:**

* The paper integrates reinforcement learning into organizational design, offering a novel alternative to traditional analytical methods. I believe this approach to be quite novel, with the potential to assist humans in making faster and more accurate decisions across diverse organizational scenarios.
* The authors identify and motivate the research problem clearly.

**Weaknesses:**

1. MOPOMDP definition. One of the paper’s main contributions is the reformulation of the organizational-design problem from a reinforcement learning (RL) perspective. While this is novel, the proposed MOPOMDP integration (which lacks references) introduces modifications to the standard framework that raise concerns about correctness, rigor, and applicability. The authors effectively force a single-agent formulation on a problem that naturally involves multiple actors across hierarchical levels, systems extensively studied in the hierarchical multi-agent RL (MARL) literature. As a result, the manuscript adopts strong, unrealistic, assumptions (e.g., actors follow discrete, deterministic policies) and leaves the composition of canonical spaces (observation, action, and state spaces) ambiguous.

Given that problem modelling is a main contribution, the proposed framework should be: (i) comprehensive, i.e., not restricted by these limiting assumptions; (ii) rigorous, with each element formally defined and motivated; and (iii) clear, so it can serve as a reliable foundation for subsequent work. I recommend the authors revise the formulation with these points in mind and connect their work with the hierarchical MARL literature when doing so.

2. Temporally extended actions treatment. The framework’s architecture consists of a team of LLM agents with distinct roles and incentives, coordinated by a management policy (also implemented as an LLM) that operates through higher-level incentives. Each lower-level LLM executes actions that span multiple time steps. However, the manner in which the authors handle these temporally extended actions, while claiming that the management policy can be queried at every time step, is unclear and requires a more detailed explanation and analysis. I note that the asynchronous execution of temporally extended actions in multi-actor systems is an active research area (see the MacDec-POMDP literature).

3. Lack of analysis of "hallucinations". The clinical trial scenario is motivated well, but the paper fails to study the innate limitations of the use of LLMs for their simulated scenarios. For example, "hallucinations" are a key challenge of LLMs and the proposed problem framework seems particularly sensitive to them. A reported 15\% hallucination rate is concerning, given that errors in such complex systems could lead to serious consequences, including financial losses or potential risks to patient safety in the clinical trial scenario. A more detailed discussion, including illustrative examples, would strengthen the paper and help clarify the practical implications of these errors while also identifying directions for future work.

4. Overall clarity. In addition to the issues described above, which contribute to a lack of clarity, multiple statements are either poorly constructed or incomplete, and pieces of information are dispersed across the paper without clear referencing. Some examples include: (i) indexing inconsistencies (line 152, the time and actor indexes are swapped; line 227, unnecessary ' for $s_{t+1}$' and missing time index for z; line 257; and multiple others throughout the main text); (ii) undefined variables (line 151, observation space O; line 158, psi); (iii) figure inaccuracies (line 278 mentions at most one outgoing edge per node but Figure 2 showcases multiple such edges); and (iv) management policy actions (Section 2.3 could be extended to explicitly specify the expected output format for each action).

**Questions:**

1. In line 149, defining the actor policies as discrete mappings seems quite restrictive, particularly in an organizational setting where complex actors (e.g., humans) may exhibit inherent stochasticity in their action choices. Could the authors elaborate on the rationale behind this design choice, which departs from the more general formulation of policies as distributions? I note that introducing stochasticity would indeed affect the transition function derivation in lines 164–165; however, this should not serve as the sole motivation for enforcing such a constraint.

2. In line 151, the text suggests that all agents share the same observation space. Given that agents have different roles, and that in real organizations various departments typically have access to distinct channels of information, could the authors elaborate on the motivation behind this modeling choice, which again seems restrictive?

3. The authors state in line 167, that the motivation for vectorial reward functions is "competing goals inherent in any complex organization", and proceed to integrate a set of metrics in the clinical trial example as this reward vector. However, it is unclear which metrics from Section 3.3 are actually used. Can the authors provide a clearer description of how this reward vector is constructed?

4. Figure 5 does not specify the values of the two parameters associated with the Fixed Plan policies (Length and Stride). Furthermore, it is unclear how the authors extended Fixed Plan policies to depend on these parameters. Could the authors provide more information about this process? Without such clarification, the figure cannot be properly interpreted or taken as a meaningful result.

---

> ### Author Response · Authors · 2025-11-21
>
> We thank this reviewer for taking the time to review our work. We have carefully considered each point of feedback and provide our point-by-point responses below. Please don’t hesitate to let us know if any further clarifications are required.
>
> **P1 MOPOMDP Formalism**
>
> We appreciate the reviewer’s careful reading and agree that our current formalization can be sharpened and better connected to the hierarchical multi-agent RL literature.
>
> a) **Why is a single-agent MOPOMDP appropriate in our setting?**
>
> Our setting is indeed naturally multi-agent and hierarchical: there is a management layer and multiple operational actors with heterogeneous roles and incentives. Conceptually, the full system can be cast as a hierarchical Dec-POMDP [1].
> In this work, however, we deliberately fix the policies of the operational actors (the LLM-based roles and their tools) and only learn or benchmark policies at the management level. Under this assumption, the lower-level actors can be absorbed into the transition dynamics of an induced decision problem faced by the manager. Formally, fixing $\{\pi_g\}$ defines a stochastic transition kernel $P(s_{t+1} \mid s_t, z_t)$ over the global state when the manager selects configurations  $z_t$. The manager’s problem is therefore a (multi-objective) POMDP over this induced environment. This reduction is now made explicit in the revised version and clarifies that we are not denying the underlying multi-agent nature, but rather working with a standard reduction obtained by fixing lower-level policies.
>
> **Action taken.** We have added a dedicated paragraph in the formalism section, positioning our model as a specific instantiation of a hierarchical Dec-POMDP. We have expanded Section 5, Appendix A, and Table 3 with hierarchical Dec-POMDP references.
>
> b) **Clarifying that actor policies are stochastic, not deterministic**
>
> The reviewer is correct that our current notation, $\pi_g: \tilde{o}_t^g, z_t^g, I_t^g \mapsto a_t^g,$ suggests a deterministic mapping. This was not intended as a modelling assumption, but as a simplification in the exposition. All results and constructions remain valid when  $\pi_g$  is interpreted as a stochastic policy. In the actual OSE implementation, actors are instantiated as LLM-based agents, and their behavior is inherently stochastic.
> In the revised paper, we adopted standard stochastic policy notation for both actors and the manager. For each actor $g$, we will use $\pi_g(a_t^g \mid \tilde{o}_t^g, z_t^g, I_t^g)$, denoting a distribution over actions in the actor’s action space $\mathcal{A}$.  All randomness from LLM sampling, tool outcomes, and mechanistic models is captured in these stochastic kernels and in the transition function $P$.
>
> **Action taken.** We have updated the formalism and text to make clear that we do not assume deterministic policies.

---

> ### Author Response · Authors · 2025-11-21
>
> c) **Making the canonical spaces explicit and rigorous**
>
> We agree that the canonical spaces were underspecified. In the revision, we provide a more standard multi-objective POMDP definition for the management policy, and we clarify the underlying multi-agent structure. Concretely, we now define a multi-objective POMDP tuple $\langle \Theta, S, \mathcal{Z}, P, \vec{R} \rangle$ for the management policy (the only learner in this work), where $\Theta$ and $\mathcal{Z}$ describe the observation and action spaces at the management level. We then use a partially separate Dec-POMDP notation $\langle O^n, S, A^n, P, R \rangle$ to describe the collection of low-level agents whose policies are fixed.
>
> Moreover, in our formalism, we define a single observation space $O$ and a single action space $A$ for the low-level actors because they operate in the same underlying environment and all observations are generated from shared channels, namely inter-actor communication and the outcomes of actions and tool calls. The global observation space therefore, represents the union of all possible observations that can be produced by these common sources.
>
> However, in practice, and as described in Section 2.2.3, our implementation imposes simplifying assumptions that restrict what each actor can effectively observe. Each actor is tied to a specific role $\gamma$ from a finite set of roles $\Gamma$, which determines its prompt and its set of tools, thereby reducing both its available actions and the information it can access. As a result, a role $\gamma$ restricts the actor’s actions and observations to subsets $A_{\gamma} \subseteq A$ and $O_{\gamma} \subseteq O$ of the global action and observation spaces. For example, the Regulatory Lead cannot see failure messages generated by analysis tools that are only available to the Statistician. Thus, although the model is written with a single observation space and a single action space, each role in the OSE interacts with a role-specific subset of these spaces. In the revised paper, the effect of the simplified assumptions on the observation and action spaces has been specified. In section 2.2.3, we now write these spaces as unions of role-specific subspaces:
> $$
> A = \bigcup_{\gamma \in \Gamma} A_{\gamma}, \qquad
> O = \bigcup_{\gamma \in \Gamma} O_{\gamma}.
> $$
>
> **Action Taken.** Section 2.2.3 of the revised manuscript has been updated with the definition of the observation space and action space.
>
> [1] (2016). *A Concise Introduction to Decentralized POMDPs*

---

> ### Author Response · Authors · 2025-11-21
>
> **P2 Temporally extended actions**
>
> We thank the reviewer for raising this point, which enables us to clarify the temporal interaction between the management policy and the actors.
>
> **a) Management policy decision frequency**
>
> In our setting, the management policy is defined, following the POMDP formalism, as a policy that is queried at every environment time step.  However, when the organization is in a state where every actor is either inactive or already engaged in temporally extended action, applying a new configuration that leaves each actor in its current state does not modify the organization’s state, as if the policy had not been queried. One of the insights of the Clinical Trial OSE is that it is important to know when to enter such waiting states (e.g., while a clinical study is running and no structural change is needed).
>
> **b) Asynchronous macro-actions across multiple actors**
>
> The reviewer rightly points out that asynchronous execution of temporally extended actions in multi-actor systems is an active research area. Our environment instantiates a special case, where each actor can engage in a temporally extended action (for instance, a tool call ), which has stochastic durations (e.g., number of simulated hours/days until completion, see Table 6 ), governed by mechanistic models and external factors. Since actors are single-threaded, while a temporally extended action is in progress, that actor cannot start another one, but different actors can be in different actions simultaneously, with different remaining durations.
>
> **c) Manager’s ability to intervene during long-running actions**
>
> A practical question is whether the management policy can interrupt or reconfigure actors while a temporally extended action is in progress. In our implementation, temporally extended actions are non-interruptible: the manager cannot modify that actor’s configuration until the action completes and the actor becomes ready again at a subsequent time step. However, the manager can still reconfigure other actors and adjust communication patterns during this period. This blocking-state design provides a simple and explicit way to integrate asynchronous, temporally extended actions into the discrete-time transition kernel, while still allowing the management policy to be formally queryable at every time step. The management policy observes the availability status of each actor through their observations and thus knows which actors are currently blocked. When applied to every agent, this assumption corresponds to the “synchrony assumption” in the Event-Driven Dec-POMDP [1]; here, only the management policy has access to each actor’s observation. Future work could relax this assumption by modeling imperfect communication between the management policy and the organization’s actors.
>
> **Action Taken.**  We updated section 2 of the revised manuscript to explicitly describe: (i) how temporally extended actions are represented as blocking states with stochastic durations; (ii) that the management policy is queried at every time step (iii) the synchrony assumption underlying the manager’s access to actors’ observations.
> ****
>
> [1] (2013) Multiagent POMDPs with asynchronous execution

---

> ### Author Response · Authors · 2025-11-21
>
> **P4 Improved Clarity**
>
> We thank the reviewer for pointing out these issues affecting clarity, and we have revised the manuscript accordingly. We carefully edited the text to improve wording and internal referencing, and to ensure that related pieces of information are clearly cross-referenced.
>
> **Actions taken.** (i) We corrected all indexing inconsistencies, including the swapped time and actor indices at line 152 and the unnecessary “for” as well as the missing time index for z at line 227, and we checked the rest of the manuscript for similar issues. (ii) We now explicitly introduce the observation space $O$ and the policy space $\Psi$ before their first use. (iii) We clarified the description of the configuration graph so that it is fully consistent with Figure 2: we now state that each node can have at most one *type* of outgoing edge. (iv) In appendix B.1 referenced in Section 2.3, we explicitly specify the expected output format for each management policy action, which in practice relies on a dictionary format with an adjacency list to communicate the configuration graph.
>
> **P5 Reward vectors**
>
> We thank the reviewer for pointing out this lack of clarity. In the Clinical Trial OSE, the reward vector uses all metrics defined in Section 3.3: Completion, Correct outcome, Total time, Worked time, Studies completed count, Patients hired count, and Adverse events count. Formally, the reward vector is simply the concatenation of these metrics for each episode, as reflected in the benchmark results reported in Section 4.1. In this work, we do not train management policies with this vector; it is only used for evaluation and for characterizing empirical trade-offs between objectives.
>
> **Action taken.** We have clarified in the Section 3.3 of the revised manuscript that the vectorial reward is constructed directly from the 7 metrics defined in this section.
>
> **P6 Fixed Plan policies**
>
> We thank the reviewer for highlighting this omission. In the revision, we report in a new appendix the exact Length and Stride values used for each Fixed Plan configuration (both in the benchmark and in the Pareto frontier experiment). This will remove ambiguity about how the Fixed Plan policies are constructed and interpreted.
>
> **Action Taken.** Details concerning the fixed plan policies are added in Appendix B.3.

---

### Meta-Review · Area_Chair_mEEh · 2026-01-06

**Summary:**

This paper proposes a simulation environment for organization design. The paper formulates this problem as multiagent decision making process and provides a concrete clinical benchmark. The paper further provides some preliminary comparisons across models. Reviewers were concerned about how ad hoc the overall formulation was as well as the limited number of benchmarks and no real world results. Since the authors did not provide new experimental results or paper edits to alleviate these concerns, the AC recommends rejection of the paper.

**Reviewer Concerns:**

Reviewers were concerned about how ad hoc the overall formulation was, as well as the limited number of benchmarks and no real world results.

**Reviewer Scores:**

Given that each reviewer recommended rejection for the paper and the limited additional experimental evidence provided by the authors, I believe that each reviewer would have maintained their scores.

---

### Decision · Program_Chairs · 2026-01-26

Reject